# Metabolic Engineering Interventions for Sustainable 2,3-Butanediol Production in Gas-Fermenting *Clostridium autoethanogenum*

Parsa Ghadermazi,[a] Angela Re,[b] Luca Ricci,[b,c] Siu Hung Joshua Chan[a]

[a]Chemical and Biological Engineering, Colorado State University, Fort Collins, Colorado, USA
[b]Centre for Sustainable Future Technologies, Fondazione Istituto Italiano di Tecnologia, Turin, Italy
[c]Department of Applied Science and Technology, Politecnico di Torino, Turin, Italy

Parsa Ghadermazi and Angela Re contributed equally to this work. Author order was determined both alphabetically and in order of increasing seniority.

**ABSTRACT** Gas fermentation provides a promising platform to turn low-cost and readily available single-carbon waste gases into commodity chemicals, such as 2,3-butanediol. *Clostridium autoethanogenum* is usually used as a robust and flexible chassis for gas fermentation. Here, we leveraged constraint-based stoichiometric modeling and kinetic ensemble modeling of the *C. autoethanogenum* metabolic network to provide a systematic *in silico* analysis of metabolic engineering interventions for 2,3-butanediol overproduction and low carbon substrate loss in dissipated $CO_2$. Our analysis allowed us to identify and to assess comparatively the expected performances for a wide range of single, double, and triple interventions. Our analysis managed to individuate bottleneck reactions in relevant metabolic pathways when suggesting intervening strategies. Besides recapitulating intuitive and/or previously attempted genetic modifications, our analysis neatly outlined that interventions—at least partially—impinging on by-products branching from acetyl coenzyme A (acetyl-CoA) and pyruvate (acetate, ethanol, amino acids) offer valuable alternatives to the interventions focusing directly on the specific branch from pyruvate to 2,3-butanediol.

**IMPORTANCE** Envisioning value chains inspired by environmental sustainability and circularity in economic models is essential to counteract the alterations in the global natural carbon cycle induced by humans. Recycling carbon-based waste gas streams into chemicals by devising gas fermentation bioprocesses mediated by acetogens of the genus *Clostridium* is one component of the solution. Carbon monoxide originates from multiple biogenic and abiogenic sources and bears a significant environmental impact. This study aims at identifying metabolic engineering interventions for increasing 2,3-butanediol production and avoiding carbon loss in $CO_2$ dissipation via *C. autoethanogenum* fermenting a substrate comprising CO and $H_2$. 2,3-Butanediol is a valuable biochemical by-product since, due to its versatility, can be transformed quite easily into chemical compounds such as butadiene, diacetyl, acetoin, and methyl ethyl ketone. These compounds are usable as building blocks to manufacture a vast range of industrially produced chemicals.

**KEYWORDS** 2, 3-butanediol, *Clostridium autoethanogenum*, circular economy, gas fermentation, metabolic engineering

Rethinking complex value chains is a key priority to achieve a modern, low-carbon, resource- and energy-efficient economy. Sustainable and carbon circular economy models are being developed to offer functionalities analogous or superior to traditional commodities with potentially lower environmental impacts, but they currently represent a very small share of the market. One component of the technological innovations necessary to attain carbon recycling can be the application of bioprocesses

Address correspondence to Angela Re, angela.re@iit.it.

The authors declare no conflict of interest.

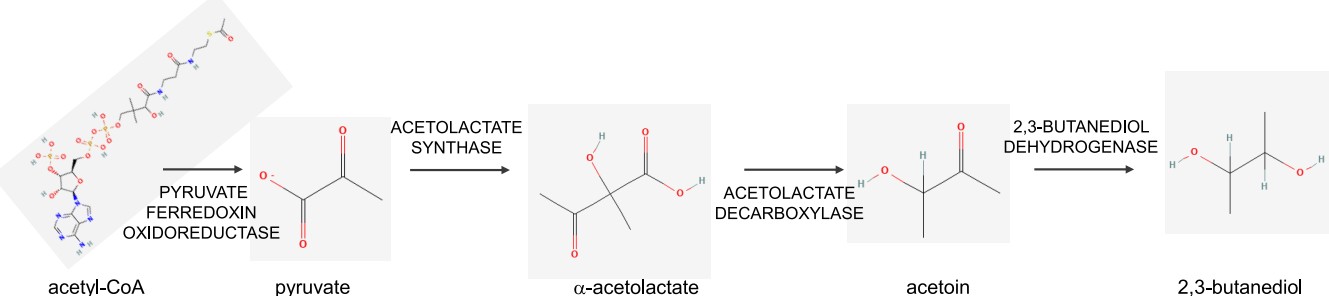

**FIG 1** 2,3-Butanediol biosynthetic pathway. Shown are the enzymes involved in the synthesis of 2,3-butanediol starting from acetyl-CoA.

enabling conversion of $C_1$-based industrial waste gas streams into commodity chemicals (1, 2). Carbon monoxide (CO) is a major by-product of the incomplete combustion of carbon-based fuels—including coal, oil, natural gas and wood. Furthermore, CO is also a major component of synthesis gas (syngas), where the relative amounts of CO and $H_2$ vary largely, depending on the feedstock, gasifier type, and process conditions adopted (3). The release of CO into the atmosphere may have a significantly negative environmental impact by affecting the abundance of greenhouse gases in the atmosphere.

The ability to grow on gaseous substrates as the sole carbon source is a defining property of the acetogens of the genus *Clostridium*, which use the carbon-fixation Wood-Ljungdahl biochemical pathway for autotrophic growth (4). Regeneration of redox cofactors occurs through reactions leading to excreted by-products. One such product of particular importance is 2,3-butanediol (2,3-BDO), a trace component of the native product profile of some $C_1$-fixing, anaerobic, acetogenic, ethanologenic, and carboxydotrophic members of the genus *Clostridium*, namely, *C. autoethanogenum*, *C. ljungdahlii*, *Clostridium* sp. strain AWRP, *C. ragsdalei*, and *C. coskatii* (5–7).

2,3-BDO is a valuable platform chemical (8) that can be used to produce butadiene (a monomer of synthetic rubber), acetoin (a volatile compound used in foods, plant growth promoters, and biological pest control), diacetyl (a flavor enhancer), and methyl ethyl ketone (an excellent organic solvent). Furthermore, hydrogenation of methyl ethyl ketone produces the fuel 2-butanol, which shows the highest octane number and the lowest boiling point among the four stereoisomers of butanol. Compared to bioethanol, butanol has a higher energy density and lower hygroscopicity. Blending of 2-butanol with gasoline does not need to modify the current vehicle system (9). 2,3-BDO can be used as antifreeze agent because of the low freezing point shown by aqueous solutions of its stereoisomers (10, 11), and it can be used as a novel chain initiator and extender in the manufacturing of polyol components of polyurethane foams (12). Furthermore, 2,3-BDO could be exploited in the cosmetics sector due to its humectant, moisturizing, and antimicrobial activities (13).

The initial reaction for the production of 2,3-BDO from acetyl coenzyme A (acetyl-CoA) is mediated by the pyruvate:ferredoxin oxidoreductase (PFOR), which is present in two copies in the genome of *C. autoethanogenum* (CAETHG_0928 and CAETHG_3029) (5, 14, 15). This enzyme catalyzes the interconversion of acetyl-CoA and $CO_2$ into pyruvate, with ferredoxin and thiamine pyrophosphate as cofactors (Fig. 1). 2,3-BDO formation from pyruvate occurs through three subsequent reactions. First, acetolactate synthase (ACLS), encoded by CAETHG_1740 in *C. autoethanogenum*, forms a molecule of acetolactate from two molecules of pyruvate, accompanied by the release of a $CO_2$ molecule. Acetolactate is then split into acetoin and $CO_2$ by acetolactate decarboxylase (ACLDC), encoded by CAETHG_2932 in *C. autoethanogenum*. Acetoin can be finally reduced to 2,3-BDO by a butanol dehydrogenase (here BTDDx), which is encoded by CAETHG_0385 in *C. autoethanogenum*, or by a primary-secondary alcohol dehydrogenase (here BTDDy), which is encoded by CAETHG_0553 in *C. autoethanogenum* (16, 17). It has been hypothesized that this primary-secondary NADPH-dependent alcohol dehydrogenase is also involved in ethanol

production and not only in 2,3-BDO synthesis (18). Whereas this enzyme strictly depends on NADPH (17), the reduction of acetoin to 2,3-BDO by BTDDx is favored by NADH, even if it has been observed with both NADPH and NADH (16).

Although 2,3-BDO production by microbial fermentation of gaseous substrates containing CO has been demonstrated (19), 2,3-BDO is usually a secondary product of gas fermentation processes. It is desirable to be able to affect the fermentation in such a way that the production of 2,3-BDO is enhanced relative to the production of other products, including ethanol, that are routinely produced in the fermentation of gaseous substrates by acetogens of the genus *Clostridium*. A few key process parameters have been identified as able to influence 2,3-BDO production. It has been shown that increasing the hydrogen composition of the gaseous substrate and the specific rate of hydrogen consumption by the microbial culture negatively impacts 2,3-BDO productivity and that increasing the age of the cells in culture favors 2,3-BDO productivity (20). Medium manipulations included providing a compound inhibiting one or more enzymes, which convert acetolactate to branched-chain amino acids to the fermentation (21), or increasing the concentration of at least one nutrient selected from the group consisting of vitamin $B_1$, vitamin $B_5$, vitamin $B_7$, and mixtures thereof above the cellular requirement of the microorganism (22). A clear metabolic engineering strategy for increased 2,3-BDO production is not available, despite the availability of genome-scale metabolic reconstructions of *C. autoethanogenum* (23–25) and the predictive engineering potential of constraint-based modeling (26), based on stoichiometry (27), and kinetic modeling, which allow us to gain insights into regulatory behaviors or rate-limiting steps (28). To avoid the hurdle of quantifying detailed enzyme kinetics of each reaction, which is impractical for large-scale networks, many frameworks explore a range of parameters and select a subset based on their consistency with experimental observations. Using the parameters obtained by training the model, the analysis of the network kinetics allows formulating potential engineering strategies. The ensemble modeling (EM) framework uses phenotypic data, such as flux changes due to changes in enzyme expression, to screen for kinetic models. The EM approach builds up an ensemble of models that span the space of kinetics allowable by thermodynamic constraints and that would all reach the given steady state, when considering flux distribution and metabolite concentrations. The constructed models can then be employed to predict system phenotypes, such as flux changes due to changes in enzymes' expression levels (29, 30). Cells are typically robust to perturbations to native enzyme concentrations. However, the addition of heterologous pathways may not result in similar steady-state stability. The robustness issue of nonnatural pathways is addressed in ensemble modeling for robustness analysis by computing the likelihood for an intervention to cause a metabolic instability (31). It is notable that dedicated software makes increasingly available the breadth of modeling techniques addressing the thermodynamic and kinetic feasibility of a metabolic pathway and its robustness against perturbations (32).

Metabolic engineering to obtain 2,3-BDO is currently limited by the fact that acetogens live at the thermodynamic edge of life (33). As shown in Fig. 2, the Wood-Ljungdahl pathway, which acts as a terminal electron-accepting/energy-conserving process and as a mechanism for $CO_2$ assimilation into cell carbon (34), is neutral with respect to ATP production via substrate-level phosphorylation (33), when acetyl-CoA is transformed into acetate (4). Consequently, autotrophic growth of acetogens of the genus *Clostridium* is strictly dependent on the chemiosmotic energy conservation process. The only coupling site for energy conservation is the Rnf complex, whereby the free energy change of the electron transport is coupled to the extrusion of ions from the cytoplasm to the periplasm. The generated electrochemical ion gradient across the membrane drives ATP synthesis via a membrane-bound $F_0F_1$-ATP synthase (35).

When using CO as the sole carbon source, in the absence of significant $H_2$, a fraction of CO is converted to $CO_2$ (36–38). The production of $CO_2$ represents inefficiency in the overall carbon capture and, if released, $CO_2$ has the potential to contribute to greenhouse gas emissions.

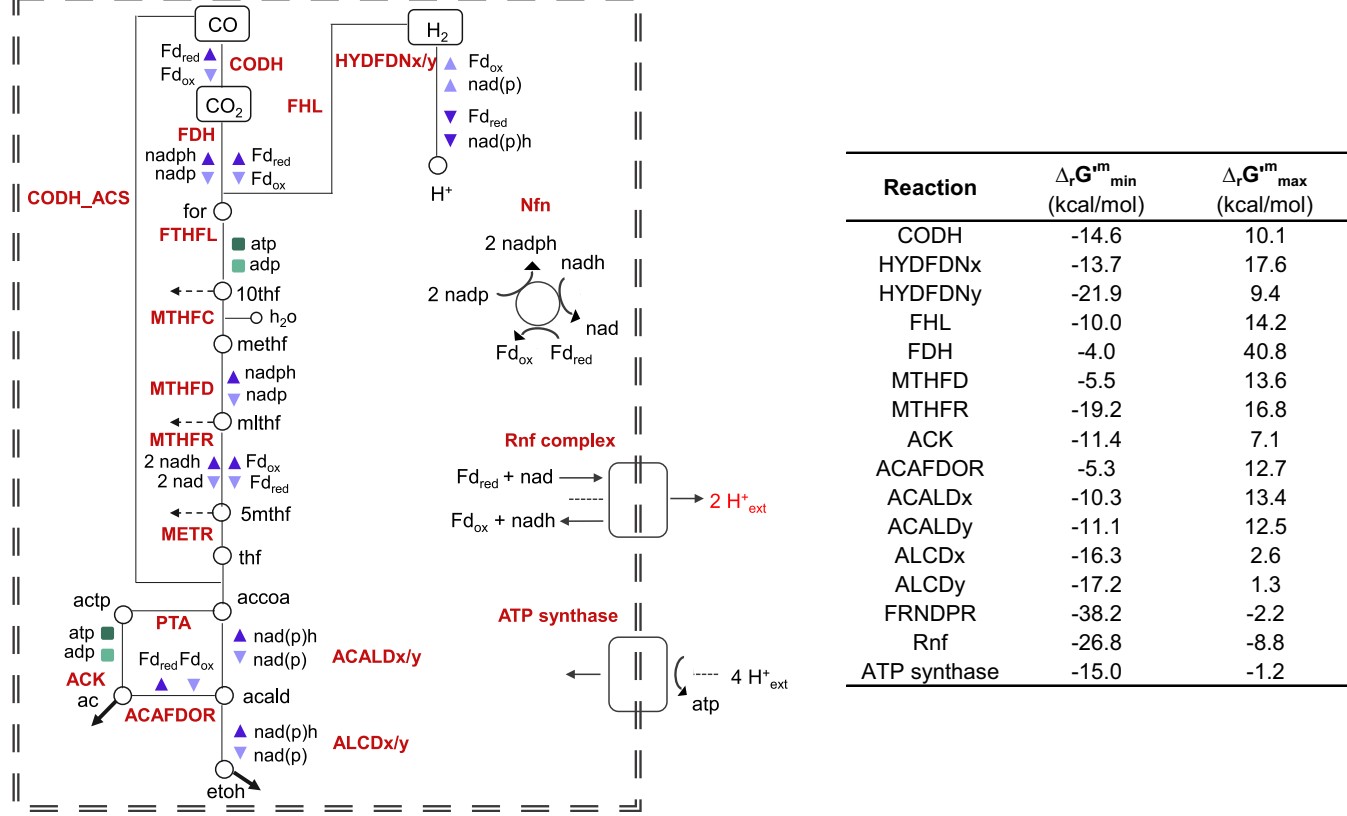

| Reaction | $\Delta_r \mathbf{G'}^m_{min}$ (kcal/mol) | $\Delta_r \mathbf{G'}^m_{max}$ (kcal/mol) |
|---|---|---|
| CODH | -14.6 | 10.1 |
| HYDFDNx | -13.7 | 17.6 |
| HYDFDNy | -21.9 | 9.4 |
| FHL | -10.0 | 14.2 |
| FDH | -4.0 | 40.8 |
| MTHFD | -5.5 | 13.6 |
| MTHFR | -19.2 | 16.8 |
| ACK | -11.4 | 7.1 |
| ACAFDOR | -5.3 | 12.7 |
| ACALDx | -10.3 | 13.4 |
| ACALDy | -11.1 | 12.5 |
| ALCDx | -16.3 | 2.6 |
| ALCDy | -17.2 | 1.3 |
| FRNDPR | -38.2 | -2.2 |
| Rnf | -26.8 | -8.8 |
| ATP synthase | -15.0 | -1.2 |

**FIG 2** Bioenergetics in *C. autoethanogenum*. The reducing equivalents for the reductive steps during CO oxidation are provided by the carbon monoxide dehydrogenase, which reduces Fd. The electron-bifurcating and ferredoxin-dependent transhydrogenase Nfn is transferring electrons between Fd, NADH, and NADPH. The methylene-THF reductase is assumed to be electron bifurcating. Excess $Fd_{red}$ is oxidized by the Rnf complex, which reduces NAD and builds up an $H^+$ gradient. This gradient drives ATP synthesis via the $H^+$-dependent ATP synthase. The Gibbs free energy ranges for the reactions displayed in the figure were retrieved from reference 39.

This study aims at *in silico* identification of metabolic engineering interventions for increasing 2,3-BDO production and limiting the carbon loss in the form of $CO_2$ with a *C. autoethanogenum* culture fermenting a substrate comprising CO and $H_2$ by using constraint-based stoichiometric modeling and kinetic ensemble modeling.

## RESULTS AND DISCUSSION

**Model validation.** Genome-scale metabolic models (GEMs) have been previously developed for *C. autoethanogenum*, namely, iCLAU786 (33, 34), and more recently, Metaclau (35). Prior to the use of a GEM for *in silico* strain design, we developed a three-step benchmark (https://github.com/chan-csu/OptForce_Bdoh). In the first step, we looked for any internal flux after blocking all the exchange reactions. In the second step, we looked at flux through ATP hydrolysis reaction in the absence of any exchange fluxes. iCLAU786 failed both tests, while Metaclau succeeded in both cases. There was no flux-carrying reaction in the absence of exchange fluxes in Metaclau. However, there were many reactions with non-zero flux in iCLAU786, including the ATP hydrolysis reaction, which shows infeasible energy-generating cycles. Finally, we assessed the accuracy of both genome-scale metabolic reconstructions by constraining each of them with experimental substrate uptake rates and by evaluating the ability of the models to predict experimentally observed biomass and by-products' production rates. We retrieved the experimental data sets for validation purposes from research articles that explore gas fermentations using *C. autoethanogenum* grown on CO-rich gas substrates and which report quantitative data on gas uptake and production rates (Table 1). We constrained the metabolic reconstructions with the gas uptake rates

**TABLE 1** Quantitative data sets for model validation[a]

| Reference | Gas substrate | Biomass concn | Gas substrate composition | Result (mmol/$g_{CDW}$/h) for: | | | | | | $\mu$ ($h^{-1}$) |
|---|---|---|---|---|---|---|---|---|---|---|
| | | | | $q_{CO}$ | $q_{CO2}$ | $q_{H2}$ | $q_{Acet}$ | $q_{EtOH}$ | $q_{BDO}$ | |
| 24, 44 | Syngas | LBC | 50% CO, 20% $CO_2$, 20% $H_2$, 10% Ar | −18.8 | 5.65 | −12.6 | 6.35 | 1.2 | 0 | 0.04 |
| 24, 44 | Syngas | MBC | 50% CO, 20% $CO_2$, 20% $H_2$, 10% Ar | −24.6 | 8.7 | −12.5 | 5.2 | 2.5 | 0.01 | 0.04 |
| 24, 44 | Syngas | HBC | 50% CO, 20% $CO_2$, 20% $H_2$, 10% Ar | −30.6 | 12.55 | −11.9 | 4.05 | 3.75 | 0.09 | 0.04 |
| 40 | Syngas | LBC | 50% CO, 20% $CO_2$, 20% $H_2$, 10% Ar | −18,792 | 5,625 | −12,583 | 6,333 | 1.25 | 0 | 0.04 |
| 40 | High-$H_2$ CO | LBC | 15% CO, 45% $H_2$, 40% Ar | −20,042 | 2,125 | −33,042 | 1,083 | 9,042 | 0 | 0.04 |
| 40 | CO | HBC | 60% CO, 40% Ar | −31 | 21 | 0.5 | 3.3 | 2.5 | 0.03 | 0.04 |
| 40 | Syngas | HBC | 50% CO, 20% $CO_2$, 20% $H_2$, 10% Ar | −30 | 12 | −11 | 4 | 3.75 | 0.06 | 0.04 |
| 40 | High-$H_2$ CO | HBC | 15% CO, 45% $H_2$, 40% Ar | −20 | 4 | −29 | 1.9 | 7.88 | 0 | 0.04 |

[a]Shown are the gas substrate uptake rates and production rates, which were displayed in publicly available reports on quantitative gas fermentations based on CO-rich gas substrates and reliant on *C. autoethanogenum*. The table shows the composition of the gas substrate for each experiment. LBC, MBC, and HBC represent the low, medium, and high biomass concentrations, respectively.

corresponding to each gas fermentation condition, shown in Table 1, and we carried out flux balance analysis (FBA) as well as flux variability analysis (FVA) simulations by maximizing biomass yield. The accuracy in predicting the acetate, ethanol, and 2,3-BDO production rates and the specific growth rate was found to be superior using Metaclau compared to iCLAU786. A possible reason for the better performance of Metaclau is its better model consistency since it does not contain the aforementioned energy-generating cycles. Based on the assessment of Metaclau prediction accuracy by FBA and FVA simulations (Fig. 3), we used the Metaclau GEM in the following *in silico* analysis.

**Analysis of the effect of $H_2$ on 2,3-butanediol and $CO_2$ production.** CO represents a carbon source that provides reducing equivalents via CO oxidation through a water gas shift reaction catalyzed by carbon monoxide dehydrogenase (CODH). However, the usage of CO as an electron donor is associated with the release of $CO_2$ (36–38). For example, when using only CO for 2,3-BDO production, almost two-thirds of the carbon is lost to $CO_2$ according to 11 CO + 5 $H_2O \rightarrow C_4H_{10}O_2$ + 7 $CO_2$.

Since we were interested in identifying metabolic engineering strategies to improve 2,3-BDO production in a gas feeding condition minimizing $CO_2$ production, we explored feed gas substrates combining CO with $H_2$. Indeed, if $H_2$ is present in the gas, additional reducing equivalents are made available by hydrogen oxidation ($H_2 \rightarrow 2$ $H^+$ + $2e^-$) and less CO has to be dissipated into $CO_2$ via the biological water gas shift catalyzed by CODH. Furthermore, $CO_2$ can be advantageously fixed in the Wood-Ljungdahl pathway since the direct use of $H_2$ in the $CO_2$ reduction to formate by the formate-hydrogen lyase activity (16) allows the saving of redox compared to growth on CO alone. A high-$H_2$/CO gas fermentation process using chemostat cultures of *C. autoethanogenum* (40) observed the aforementioned trend, whereby the specific $CO_2$ production rate dropped more than 5-fold by increasing the supplied $H_2$. Our FVA simulations predicted a significant drop of $CO_2$ production rate from 21 mmol/g cell dry weight [$g_{CDW}$]/h, when CO was the sole gas substrate in the Metaclau GEM, to 2 mmol/$g_{CDW}$/h, when the Metaclau GEM was constrained, with the CO and $H_2$ uptake rates corresponding to the high-$H_2$/CO gas feeding adopted in reference 40. Running FVA simulations for different combinations of CO and $H_2$ uptake rates allowed us to identify a subspace of gas uptake rates' combinations that ensure 2,3-BDO production and that substantially decrease $CO_2$ production (see Fig. S1 in the supplemental material). We ran another simulation using the kinetic ensemble model of the *C. autoethanogenum* core metabolism derived in reference 39, where we held the CO uptake constant and we varied the $H_2$ uptake rate. We did observe a decrease in the 2,3-BDO yield per hydrogen uptake. Especially after the uptake rate of hydrogen was >20 mmol/$g_{CDW}$/h, no increase in 2,3-BDO or any other significant product profile change was observed (see Fig. S2 in the supplemental material). The additional electrons at high hydrogen uptake are consumed in some futile cycles in the model. The primary reasons for this model behavior are that all kinetics in the model essentially follow Michaelis-Menten kinetics and the hydrogen-consuming reactions that lead to product formation

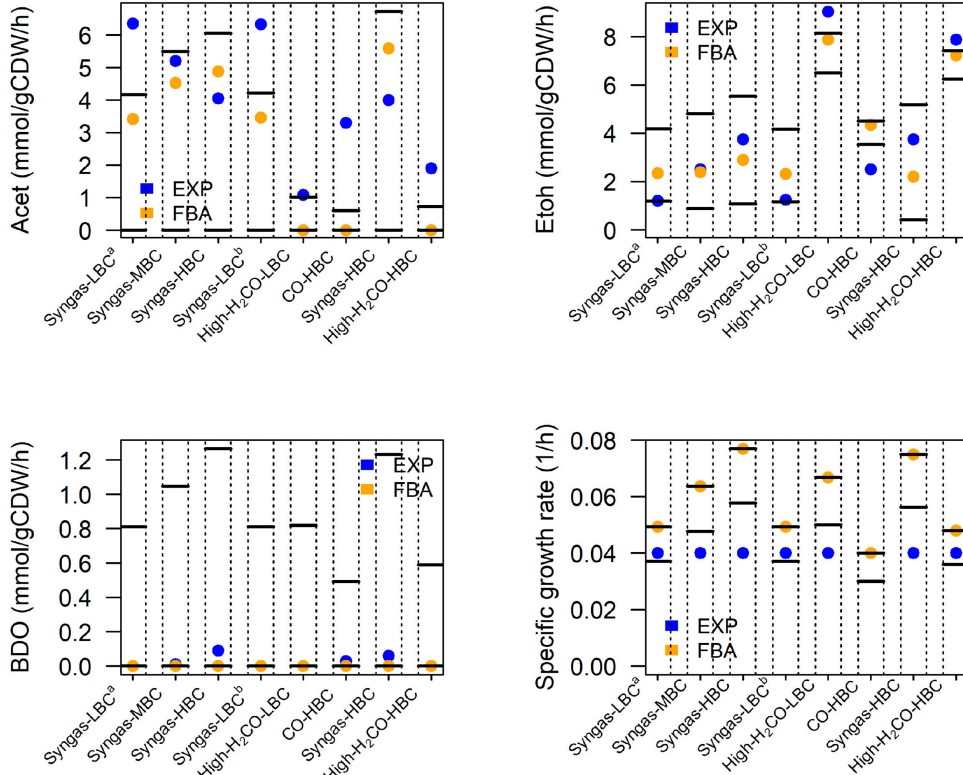

**FIG 3** Validation of Metaclau predictive capability by contrasting FBA and FVA predictions with experimentally determined production rates in benchmarking data sets. The plots show the experimental acetate, ethanol, and 2,3-BDO production rates and specific growth rate (blue dots), which are reported in Table 1, with the flux predictions obtained by FBA (orange dots) and with the flux ranges that were obtained by FVA simulations using the Metaclau GEM, constrained with the experimental gas uptake rates in Table 1.

will ultimately get saturated at a certain hydrogen concentration that is correlated with the $H_2$ uptake rate. In the light of this behavior at higher simulated $H_2$ uptakes, we constrained the GEM with the particular combination of CO and $H_2$ uptake rates reported in reference 40, which have been experimentally shown to be associated with a low $CO_2$ production rate, in order to computationally predict metabolic interventions enhancing 2,3-BDO production rate at a low $CO_2$ production rate.

**Prediction of genetic manipulations leading to 2,3-butanediol overproduction.** The OptForce framework (41) implemented within the COBRA Toolbox (42) in the MATLAB environment was used with the Metaclau GEM to enumerate the reactions that should be actively forced through genetic interventions in order to achieve the overproduction of 2,3-BDO in *C. autoethanogenum* fed with CO and $H_2$. as reported in DATA SET S1 at GitHub (https://github.com/chan-csu/OptForce_Bdoh/blob/59a42aca a7be018702cd748a441a6076391306c6/Results/Supplementary_File1.xlsx). Each intervention can foresee the increase, decrease, or elimination of the flux value corresponding to each of the involved reactions. The OptForce tool suggests genetic manipulations at the metabolic flux level. However, the lack of a quantitative mapping between flux and gene expression levels does not allow translation of the suggested sets of reaction flux changes into executable genetic modifications, especially when attempting combinatorial changes. Furthermore, the OptForce tool does not account for kinetic and thermodynamics features governing the metabolic behavior of *C. autoethanogenum* (18, 43). Therefore, it is advisable to analyze the outcomes of the reaction flux manipulations suggested by OptForce by using kinetic modeling. To this aim, we used a kinetic representation of *C. autoethanogenum* core metabolism (Fig. 4), which proved able to accurately resolve experimentally observed trends (34) and which was

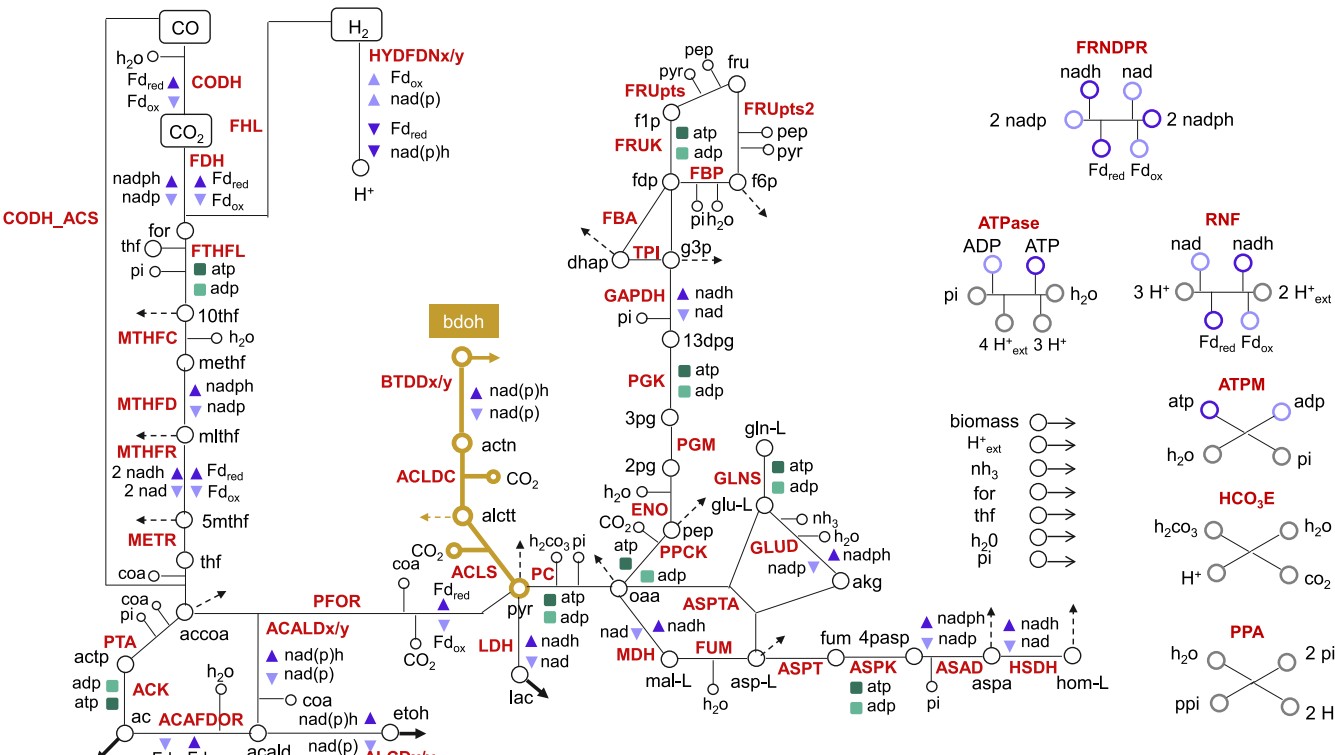

**FIG 4** *C. autoethanogenum* core metabolism. Shown is the metabolic network representative of the core metabolism of *C. autoethanogenum*. Cofactors and energy equivalents are coded, respectively, in blue and green colors. Bold arrows denote by-products' exchange reactions and dashed arrows denote metabolites included in the biomass equation. Reactions along with enzyme and metabolite abbreviations are described in Table S1.

obtained within the ensemble modeling framework (35, 36). This framework does not require the existence neither of metabolomics data nor of enzymatic expression data, but builds upon a reference steady-state flux distribution to constrain the kinetic parameters' sampling space to a realistic space and, thus, to derive the initial ensemble of multiple kinetic parameter sets. This ensemble is subsequently pruned down using perturbation data sets, which typically consist of phenotypic responses to enzymatic modifications. In reference 39, the screening of the ensemble of kinetic parametric sets did not rely on genetic perturbations, for which available data are currently limited in *C. autoethanogenum*, but on perturbations of environmental conditions. Indeed, the construction of the kinetic model relied on a public data set where gas uptake fluxes changed by effect of increasing the biomass concentration over three conditions (44). Eighteen sets of locally stable kinetic parameters fit the experimental data under all three conditions. We used this ensemble model consisting of the 18 sets of parameters for our simulations.

The kinetic ensemble model of *C. autoethanogenum* was instrumental to elucidate the change in 2,3-BDO production rate as a function of changing the expression levels of the enzymes catalyzing the reactions, which took part in the interventions suggested by OptForce. In particular, we simulated multiple scenarios where the fold changes of the enzymes catalyzing each reaction of the single, double or triple interventions were set to low ($\pm$20%), medium ($\pm$40%), or high ($\pm$60%). Each intervention is associated with a distribution of 2,3-BDO production rates that resulted from simulating all of the 18 sets of kinetic parameters of the ensemble model (39). The distributions of the simulated 2,3-BDO production rates relative to each single, double, and triple intervention are shown, respectively, in Fig. 5, Fig. 6, and Fig. 7. We then identified the interventions that resulted in significantly higher production of 2,3-BDO compared to the wild-type condition (Wilcoxon's test, significance level at 1%). Table 2, Table 3, and Table 4 report the reactions involved in statistically significant single, double, or

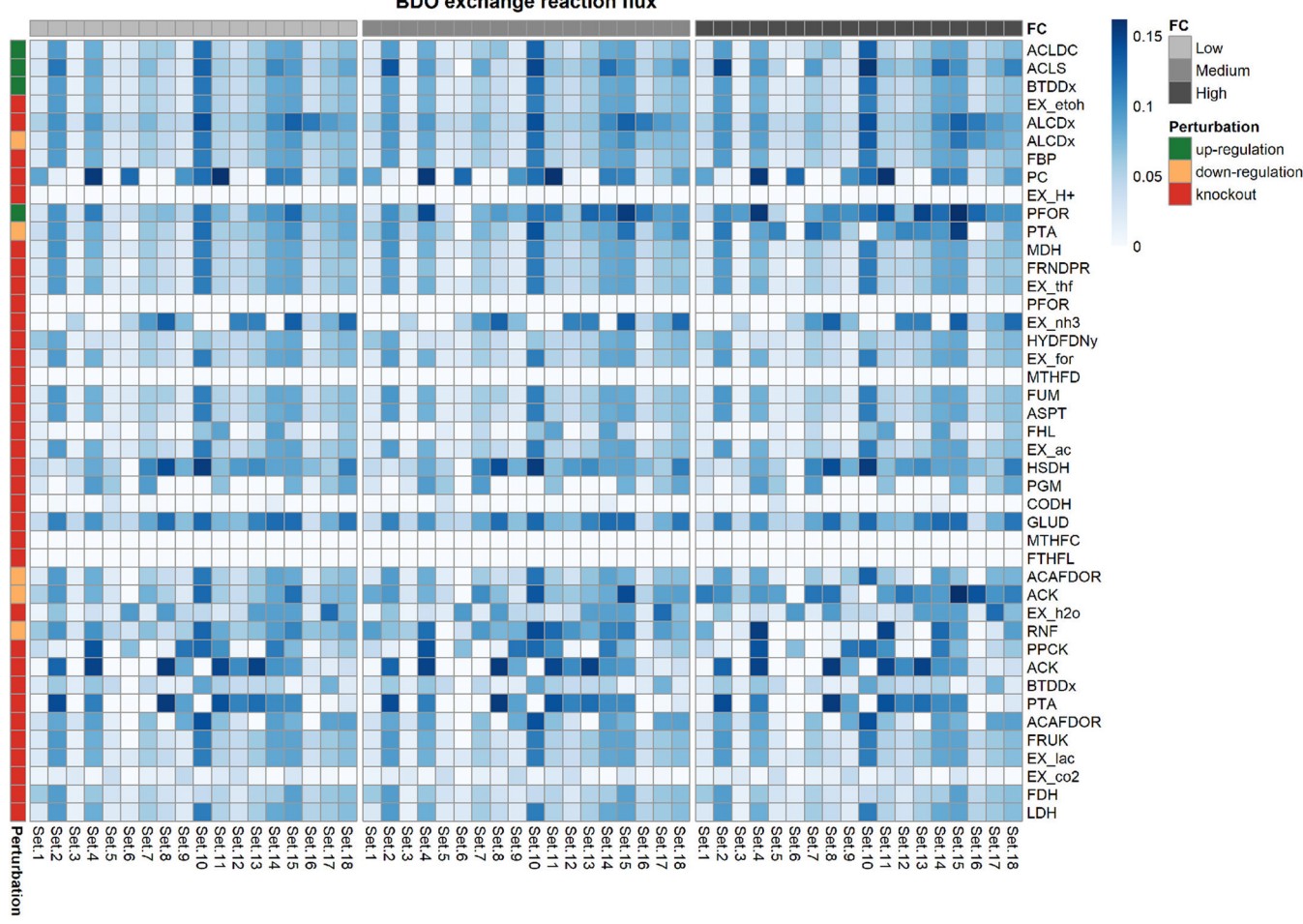

**FIG 5** Heat map showing the 2,3-BDO production rate when each of the single interventions suggested by OptForce are applied to the kinetic core model. Each row in the heat map represents an intervention and each column represents one of the 18 kinetic parameters sets. The intervention can be an upregulation, a downregulation, or a knockout. The heat map is column-wise subdivided in three blocks. Each block corresponds to different extents in the change of the levels of the enzymes catalyzing each reaction. We refer to the fold changes in enzymes' expression levels as to low, medium and high, respectively. Single interventions produced by Optforce are reported in DATA SET S1.

triple interventions, respectively. We also tested the effects of larger perturbations. We repeated the simulations to perturb the enzyme levels by 3-, 5-, and 10-fold for upregulation and 0.3-, 0.2-, and 0.1-fold for downregulation (see Table S2, Table S3, and Table S4 and Fig. S3, Fig. S4, and Fig. S5 in the supplemental material). The same strategies were suggested to increase 2,3-BDO production most significantly.

**PFOR upregulation predicted to be the most effective single intervention.** First-order interventions which were predicted to increase 2,3-BDO production, compared to the wild-type (WT) condition, primarily concern the upregulation of enzymes involved in the 2,3-BDO biosynthetic pathway. As shown in Fig. 8, overexpression of pyruvate:ferredoxin oxidoreductase (PFOR), which catalyzes the interconversion between acetyl-CoA and pyruvate, was predicted to be the most effective single intervention for improving 2,3-BDO synthesis ($\text{flux}^{2,3-BDO}_{PFOR\ up,high} = 0.10 \pm 0.023$; $\text{flux}^{2,3-BDO}_{WT} = 0.039 \pm 0.0091$), followed by the up-regulation of acetolactate synthase (ACLS) ($\text{flux}^{2,3-BDO}_{ACLS\ up,low} = 0.073 \pm 0.032$). PFOR upregulation was predicted to lead to a 2.6-fold increase in 2,3-BDO production rate. This reaction has previously been identified to be the rate-limiting step in 2,3-BDO formation in *C. autoethanogenum* (45). Therein, the authors assayed the activity of the oxidoreductase enzymes involved in the Wood-Ljungdahl pathway and in the fermentation pathway to 2,3-BDO by using *C. autoethanogenum* grown autotrophically with a feeding gas composed of 2% $H_2$, 42% CO, 20% $CO_2$, and 36% $N_2$. Measurements showed at least an activity of 1.1 U/mg for the assayed reactions, whereas the PFOR rate-limiting reaction

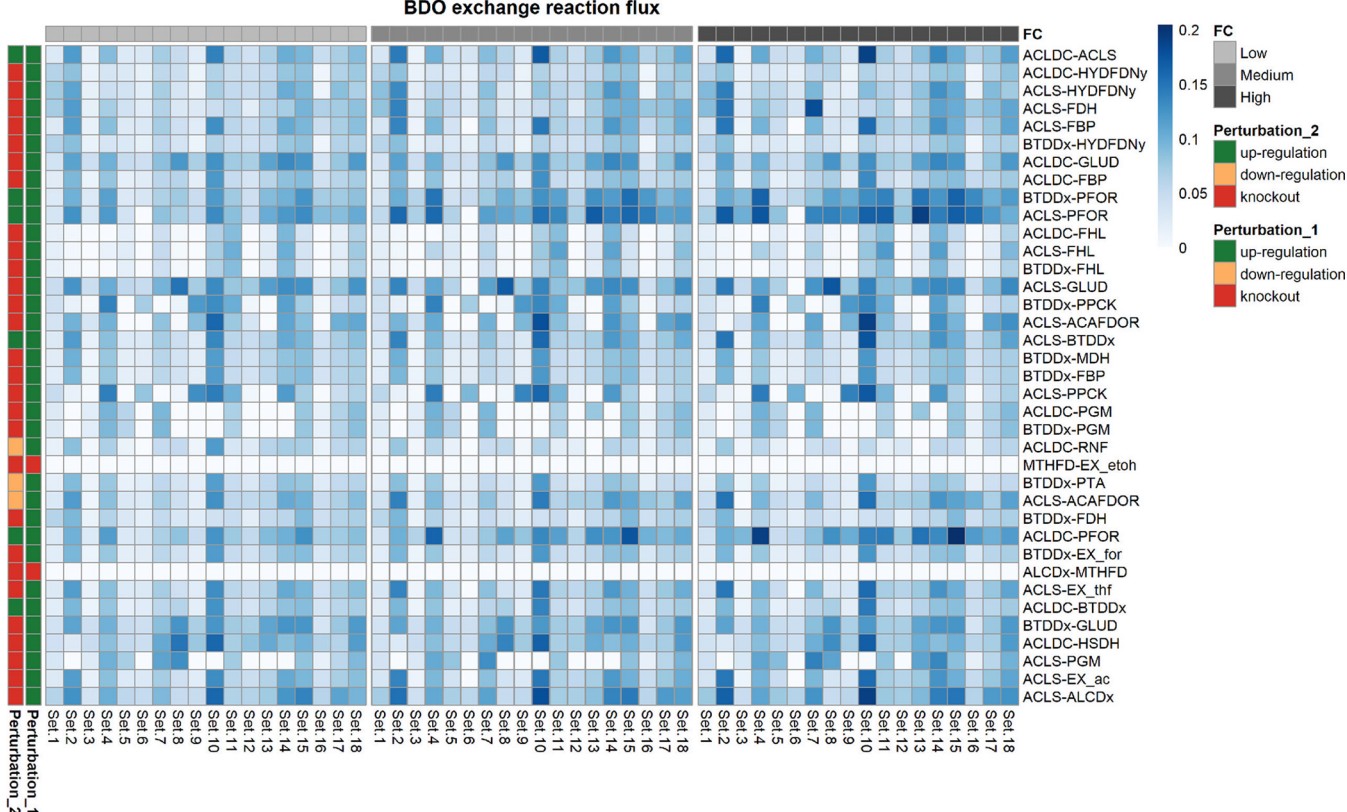

**FIG 6** Heat map showing the 2,3-BDO production rate when each of the double interventions suggested by OptForce is applied to the kinetic core model. Each row in the heat map represents a double intervention, and each column represents one of the 18 kinetic parameter sets. The perturbation applied to each reaction included in a double intervention can be an upregulation, a downregulation, or a knockout. The heat map is column-wise subdivided in three blocks. Each block corresponds to different extents in the change of the levels of the enzymes catalyzing each reaction of the double intervention. We refer to the fold changes in the enzymes' expression levels as low, medium, and high, respectively. Double interventions produced by Optforce are reported in DATA SET S1.

exhibited an enzyme activity of only 0.11 U/mg in the presence of ferredoxin, corresponding to 90% less than all other reactions assayed. PFOR overexpression was found to increase the flux through pyruvate and to increase 2,3-BDO production (45). This indicates that the ensemble kinetic model did capture some kinetic information beyond the stoichiometric model that is consistent with experimental results.

As shown in Fig. 8, an alternative kind of intervention to increase the 2,3-BDO production acts on the pathway leading to acetate and, ultimately, ethanol formation. Indeed,

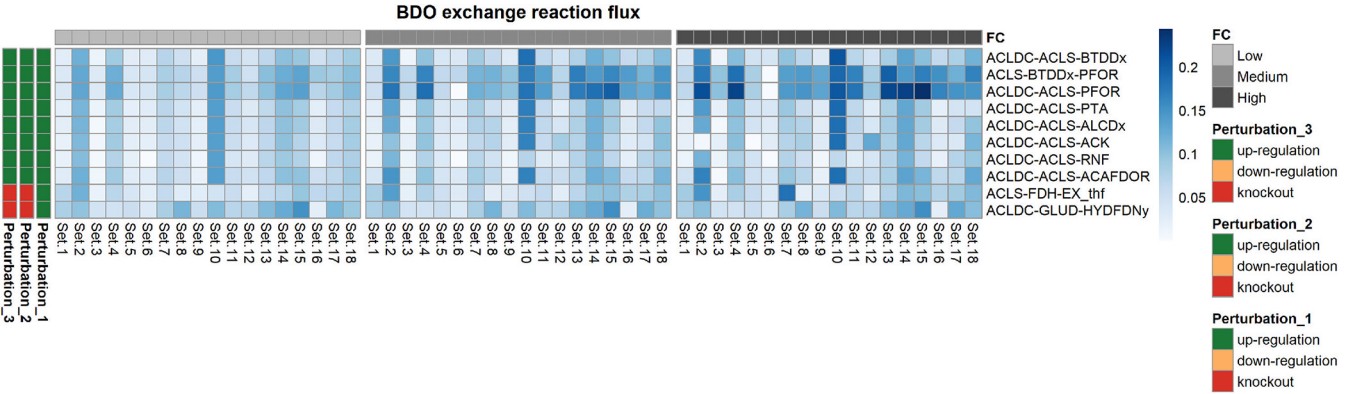

**FIG 7** Heat map showing the 2,3-BDO production rate when each of the triple interventions suggested by OptForce is applied to the kinetic core model. Each row in the heat map represents a triple intervention, and each column represents one of the 18 kinetic parameter sets. The perturbation applied to each reaction included in a triple intervention can be an upregulation, a downregulation, or a knockout. The heat map is column-wise subdivided in three blocks. Each block corresponds to different extents to the change in the levels of the enzymes catalyzing each reaction of the triple intervention. We refer to the fold changes in the enzymes' expression levels as low, medium, and high, respectively. Triple interventions produced by Optforce are reported in DATA SET S1.

**TABLE 2** Summary of first-order interventions validated by the kinetic model of *C. autoethanogenum*

| Reaction | Intervention | Adjusted *P* value | | |
|---|---|---|---|---|
| | | Low | Medium | High |
| ALCDx: $nadh + h + acald \rightleftarrows nad + etoh$ | Knockout | 1.64E−04 | 1.64E−04 | 1.64E−04 |
| PFOR: $coa + pyr + Fd\_ox \rightleftarrows co2 + accoa + h + Fd\_red$ | Upregulation | 1.64E−03 | 1.64E−03 | 1.64E−04 |
| GLUD: $h2o + nadp + glu\text{-}L \rightleftarrows nadph + nh3 + akg + h$ | Knockout | 1.64E−04 | 1.64E−04 | 1.64E−04 |
| ALCDx: $nadh + h + acald \rightleftarrows nad + etoh$ | Downregulation | 5.41E−03 | 2.30E−03 | 4.92E−04 |
| ACK: $adp + actp \rightleftarrows atp + ac + h$ | Downregulation | 8.20E−04 | 4.92E−04 | 4.92E−04 |
| ACLS: $2 pyr + h \rightleftarrows co2 + acltt$ | Upregulation | 4.10E−03 | 2.30E−03 | 8.20E−04 |
| HSDH: $nad + hom\text{-}L \rightleftarrows nadh + h + aspsa$ | Knockout | 4.10E−03 | 4.10E−03 | 4.10E−03 |
| HYDFDNy: $nadp + Fd\_ox + 2 h2 \rightleftarrows nadph + 3 h + Fd\_red$ | Knockout | 7.05E−03 | 7.05E−03 | 7.05E−03 |
| PTA: $pi + accoa + h \rightleftarrows coa + actp$ | Downregulation | 1.64E−03 | 1.64E−04 | 7.23E−02 |
| RNF: $nad + 3 h + Fd\_red \rightleftarrows nadh + 2 h\_ext + Fd\_ox$ | Downregulation | 1.64E−04 | 2.30E−03 | 1.00E+00 |

Shown are the single interventions that result in a statistically significant higher 2,3-BDO production rate compared to the wild-type condition (Wilcoxon's test, significance level at 1%).

the knockout or downregulation of ethanol:NAD oxidoreductase (ALCDx), the phosphate acetyltransferase (PTA) downregulation, and the acetate kinase (ACK) downregulation were predicted to increase 2,3-BDO production ($flux^{2,3-BDO}_{ALCDx\ KO} = 0.070 \pm 0.025$; $flux^{2,3-BDO}_{ALCDx\ down,\ high} = 0.066 \pm 0.024$; $flux^{2,3-BDO}_{PTA\ down,\ medium} = 0.087 \pm 0.030$; $flux^{2,3-BDO}_{ACK\ down,\ high} = 0.098 \pm 0.017$). PTA and ACK downregulations were predicted to result, respectively, in 2.23-fold and 2.51-fold increases in the 2,3-BDO production rate. Upon PTA and ACK downregulation, the flux through acetaldehyde:ferredoxin oxidoreductase (ACAFDOR) decreases, respectively, by 3.89-fold and by more than 5-fold compared to the wild-type case, thus limiting the consumption of reduced ferredoxin, which in turn causes the NADP to be regenerated (reduced) to NAD(P)H. The latter builds an excess that must be relieved to equilibrium and, in doing so, reduces acetoin to 2,3-BDO. Furthermore, the slight increase in 2,3-BDO production rate could be due to the flux decrease, which is generally observed through the gluconeogenesis pathway, upon both PTA and ACK downregulation. PTA downregulation negatively affects also the incomplete tricarboxylic acid (TCA) cycle. Interestingly, PTA and ACK downregulations were among the rare cases

**TABLE 3** Summary of second-order interventions validated by the kinetic model of *C. autoethanogenum*[a]

| 1st reaction | 1st intervention | 2nd reaction | 2nd intervention | Adjusted *P* value | | |
|---|---|---|---|---|---|---|
| | | | | Low | Medium | High |
| ACLS: $2 pyr + h \rightleftarrows co2 + acltt$ | Upregulation | PFOR: $coa + pyr + Fd\_ox \rightleftarrows co2 + accoa + h + Fd\_red$ | Upregulation | 1.41E−03 | 1.41E−04 | 1.41E−04 |
| ACLDC: $h + alctt \rightleftarrows co2 + actn$ | Upregulation | PFOR: $coa + pyr + Fd\_ox \rightleftarrows co2 + accoa + h + Fd\_red$ | Upregulation | 7.06E−04 | 2.68E−03 | 1.41E−04 |
| ACLS: $2 pyr + h \rightleftarrows co2 + acltt$ | Upregulation | ALCDx: $nadh + h + acald \rightleftarrows nad + etoh$ | Knockout | 1.41E−04 | 1.41E−04 | 1.41E−04 |
| ACLDC: $h + alctt \rightleftarrows co2 + actn$ | Upregulation | ACLS: $2 pyr + h \rightleftarrows co2 + acltt$ | Upregulation | 1.98E−03 | 1.41E−03 | 2.82E−04 |
| ACLS: $2 pyr + h \rightleftarrows co2 + acltt$ | Upregulation | HYDFDNy: $nadp + Fd\_ox + 2 h2 \rightleftarrows nadph + 3 h + Fd\_red$ | Knockout | 7.06E−04 | 2.82E−04 | 2.82E−04 |
| ACLDC: $h + alctt \rightleftarrows co2 + actn$ | Upregulation | GLUD: $h2o + nadp + glu\text{-}L \rightleftarrows nadph + nh3 + akg + h$ | Knockout | 2.82E−04 | 2.82E−04 | 2.82E−04 |
| BTDDx: $nad + bdoh \rightleftarrows nadh + h + actn$ | Upregulation | PFOR: $coa + pyr + Fd\_ox \rightleftarrows co2 + accoa + h + Fd\_red$ | Upregulation | 1.41E−03 | 1.98E−03 | 2.82E−04 |
| ACLS: $2 pyr + h \rightleftarrows co2 + acltt$ | Upregulation | GLUD: $h2o + nadp + glu\text{-}L \rightleftarrows nadph + nh3 + akg + h$ | Knockout | 1.41E−04 | 7.06E−04 | 2.82E−04 |
| ACLS: $2 pyr + h \rightleftarrows co2 + acltt$ | Upregulation | BTDDx: $nad + bdoh \rightleftarrows nadh + h + actn$ | Upregulation | 4.66E−03 | 1.98E−03 | 2.82E−04 |
| ACLS: $2 pyr + h \rightleftarrows co2 + acltt$ | Upregulation | ACAFDOR: $ac + 3 h + Fd\_red \rightleftarrows h2o + acald + Fd\_ox$ | Downregulation | 2.68E−03 | 7.06E−04 | 2.82E−04 |
| BTDDx: $nad + bdoh \rightleftarrows nadh + h + actn$ | Upregulation | GLUD: $h2o + nadp + glu\text{-}L \rightleftarrows nadph + nh3 + akg + h$ | Knockout | 2.82E−04 | 1.41E−03 | 2.82E−04 |
| ACLS: $2 pyr + h \rightleftarrows co2 + acltt$ | Upregulation | FDH: $nadph + 2 co2 + h + Fd\_red \rightleftarrows nadp + 2 for + Fd\_ox$ | Knockout | 2.68E−03 | 2.68E−03 | 7.06E−04 |
| ACLS: $2 pyr + h \rightleftarrows co2 + acltt$ | Upregulation | FBP: $h2o + fdp \rightleftarrows pi + h + f6p$ | Knockout | 3.53E−03 | 3.53E−03 | 7.06E−04 |
| ACLS: $2 pyr + h \rightleftarrows co2 + acltt$ | Upregulation | EX_thf: $thf\rightarrow$ | Knockout | 3.53E−03 | 3.53E−03 | 7.06E−04 |
| ACLS: $2 pyr + h \rightleftarrows co2 + acltt$ | Upregulation | EX_ac: $ac\_ext\rightarrow$ | Knockout | 3.53E−03 | 3.53E−03 | 7.06E−04 |
| ACLDC: $h + alctt \rightleftarrows co2 + actn$ | Upregulation | HSDH: $nad + hom\text{-}L \rightleftarrows nadh + h + aspsa$ | Knockout | 1.98E−03 | 4.66E−03 | 2.68E−03 |
| ACLDC: $h + alctt \rightleftarrows co2 + actn$ | Upregulation | HYDFDNy: $nadp + Fd\_ox + 2 h2 \rightleftarrows nadph + 3 h + Fd\_red$ | Knockout | 4.66E−03 | 4.66E−03 | 3.53E−03 |
| BTDDx: $nad + bdoh \rightleftarrows nadh + h + actn$ | Upregulation | HYDFDNy: $nadp + Fd\_ox + 2 h2 \rightleftarrows nadph + 3 h + Fd\_red$ | Knockout | 9.88E−03 | 9.88E−03 | 9.88E−03 |

[a]Shown are the double interventions that result in a statistically significant higher 2,3-BDO production rate compared to the WT (Wilcoxon's test, significance level at 1%).

**TABLE 4** Summary of third-order interventions validated by the kinetic model of *C. autoethanogenum*[a]

| 1st reaction | 1st intervention | 2nd reaction | 2nd intervention | 3rd reaction | 3rd intervention | Adjusted P value | | |
|---|---|---|---|---|---|---|---|---|
| | | | | | | Low | Medium | High |
| ACLS: 2 pyr + h ⇌ co2 + acltt | Upregulation | BTDDx: nad + bdoh ⇌ nadh + h + actn | Upregulation | PFOR: coa + pyr + Fd_ox ⇌ co2 + accoa + h + Fd_red | Upregulation | 1.91E−04 | 3.81E−05 | 3.81E−05 |
| ACLDC: h + alctt ⇌ co2 + actn | Upregulation | ACLS: 2 pyr + h ⇌ co2 + acltt | Upregulation | PFOR: coa + pyr + Fd_ox ⇌ co2 + accoa + h + Fd_red | Upregulation | 1.91E−04 | 1.91E−04 | 3.81E−05 |
| ACLDC: h + alctt ⇌ co2 + actn | Upregulation | GLUD: h2o + nadp + glu-L ⇌ nadph + nh3 + akg + h | Knockout | HYDFDNy: nadp + Fd_ox + 2 h2 ⇌ nadph + 3 h + Fd_red | Knockout | 3.81E−05 | 1.91E−04 | 3.81E−05 |
| ACLDC: h + alctt ⇌ co2 + actn | Upregulation | ACLS: 2 pyr + h ⇌ co2 + acltt | Upregulation | BTDDx: nad + bdoh ⇌ nadh + h + actn | Upregulation | 7.25E−04 | 7.25E−04 | 7.63E−05 |
| ACLDC: h + alctt ⇌ co2 + actn | Upregulation | ACLS: 2 pyr + h ⇌ co2 + acltt | Upregulation | ACAFDOR: ac + 3 h + Fd_red ⇌ h2o + acald + Fd_ox | Upregulation | 7.25E−04 | 7.25E−04 | 7.63E−05 |
| ACLS: 2 pyr + h ⇌ co2 + acltt | Upregulation | FDH: nadph + 2 co2 + h + Fd_red ⇌ nadp + 2 for + Fd_ox | Knockout | EX_thf: thf→ | Knockout | 7.25E−04 | 7.25E−04 | 1.91E−04 |
| ACLDC: h + alctt ⇌ co2 + actn | Upregulation | ACLS: 2 pyr + h ⇌ co2 + acltt | Upregulation | ALCDx: nadh + h + acald ⇌ nad + etoh | Upregulation | 1.26E−03 | 1.26E−03 | 2.67E−04 |
| ACLDC: h + alctt ⇌ co2 + actn | Upregulation | ACLS: 2 pyr + h ⇌ co2 + acltt | Upregulation | PTA: pi + accoa + h ⇌ coa + actp | Upregulation | 2.67E−03 | 6.45E−03 | 3.36E−03 |
| ACLDC: h + alctt ⇌ co2 + actn | Upregulation | ACLS: 2 pyr + h ⇌ co2 + acltt | Upregulation | RNF: nad + 3 h + Fd_red ⇌ nadh + 2 h_ext + Fd_ox | Upregulation | 2.00E−02 | 2.00E−02 | 6.45E−03 |
| ACLDC: h + alctt ⇌ co2 + actn | Upregulation | ACLS: 2 pyr + h ⇌ co2 + acltt | Upregulation | ACK: adp + actp ⇌ atp + ac + h | Upregulation | 6.45E−03 | 2.67E−03 | 5.20E−02 |

[a]Shown are the triple interventions that result in a statistically significant higher 2,3-BDO production rate compared to the WT (Wilcoxon's test, significance level at 1%).

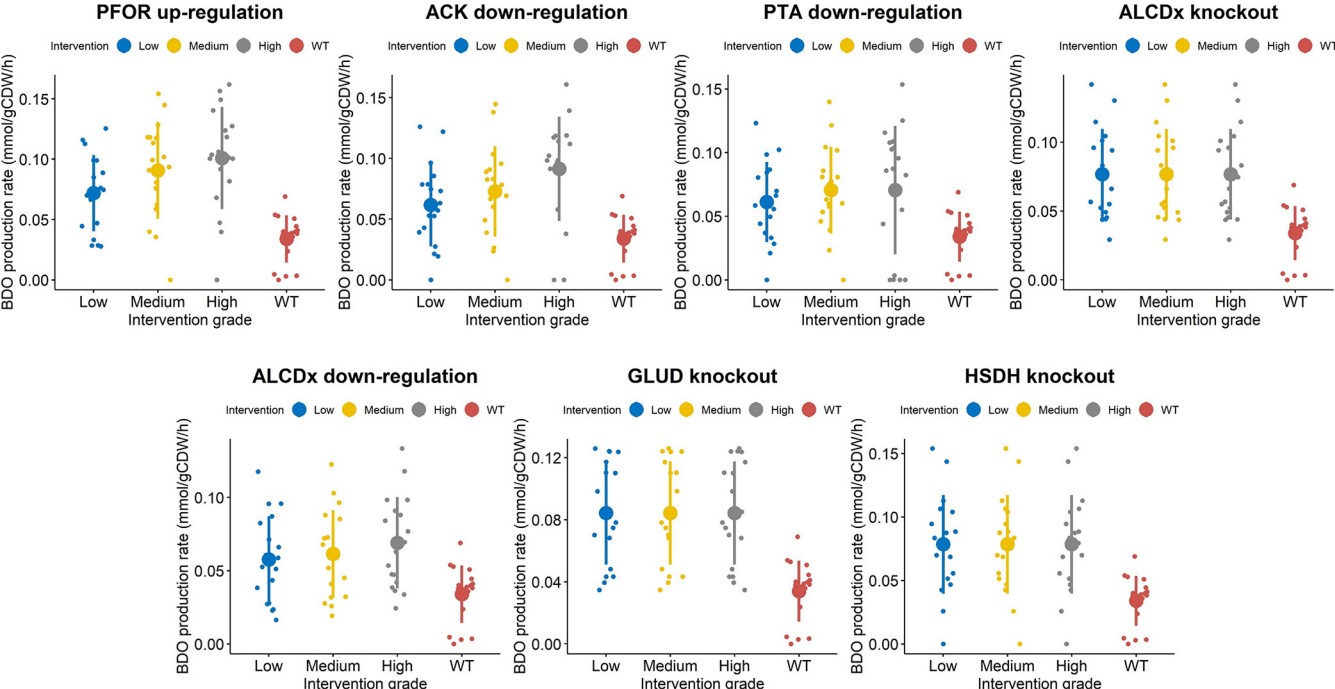

**FIG 8** Plots showing the distribution of 2,3-BDO production rates across 18 kinetic parametric sets, under the wild-type scenario and under the single-modification scenario. For each intervention, three (low, medium, and high) in the level of the enzyme catalyzing each reaction were simulated. Overlaid are the mean and standard deviation of the 2,3-BDO production rates corresponding to each intervention at each simulated change in gene expression. The figure only refers to selected single interventions reported in Table 2.

where the level of change in enzyme expression was predicted by the kinetic ensemble model to influence the effect size on 2,3-BDO production, with only a moderate phosphate acetyltransferase and acetate kinase downregulation found effective. This outcome could be explainable by the key role of the conversion of acetyl-CoA to acetate via acetyl-phosphate for ATP production in the energy metabolism of acetogens (46). This also highlights the importance of an accurate quantitative model for effective optimization of gene expression. Unless differently specified, here we report the 2,3-BDO production rates, which were estimated by simulating the highest change in the expression levels of the enzymes involved in each intervention.

The benefit induced by the knockout of ALCDx, which catalyzes the NAD-dependent conversion of acetaldehyde into ethanol, is probably attributable to the negation of a competitive reaction for the electrons required for 2,3-BDO in the form of NADH. Indeed, the flux through the NAD-dependent butanediol dehydrogenase increases 29%. Furthermore, the ALCDx knockout induces a 2.78-fold-higher flux through the NADP-dependent acetaldehyde dehydrogenase (ACALDy), which operating in the opposite direction relative to ethanol formation, increases the acetyl-CoA pool for PFOR whose flux increases 14%.

**Overexpression of both PFOR and ACLS predicted as the most effective double intervention.** Enhancing the expression of multiple enzymes involved in the 2,3-BDO biosynthetic pathway appears as a relevant choice in double interventions (Fig. 9). Many of the double interventions rely on the enzymes highlighted in the analysis of single interventions and propose the combined intervention on two of such enzymes. Notably, our analysis revealed that the preferable enzyme to overexpress along with PFOR is ACLS, which lies immediately downstream to pyruvate oxidoreductase and converts pyruvate to acetolactate and carbon dioxide. This double intervention resulted in a 1.4-fold-higher 2,3-BDO production rate compared to overexpression of only the pyruvate oxidoreductase gene, with a 3.59-fold-higher 2,3-BDO production rate compared to the wild-type condition (flux$_{\text{ACLS up,PFOR up}}^{2,3-\text{BDO}}$ = 0.14±0.030). On the contrary, overexpression of both PFOR and BTDDx or PFOR and ACLDC resulted in an almost identical 2,3-BDO production rate compared with just overexpression of

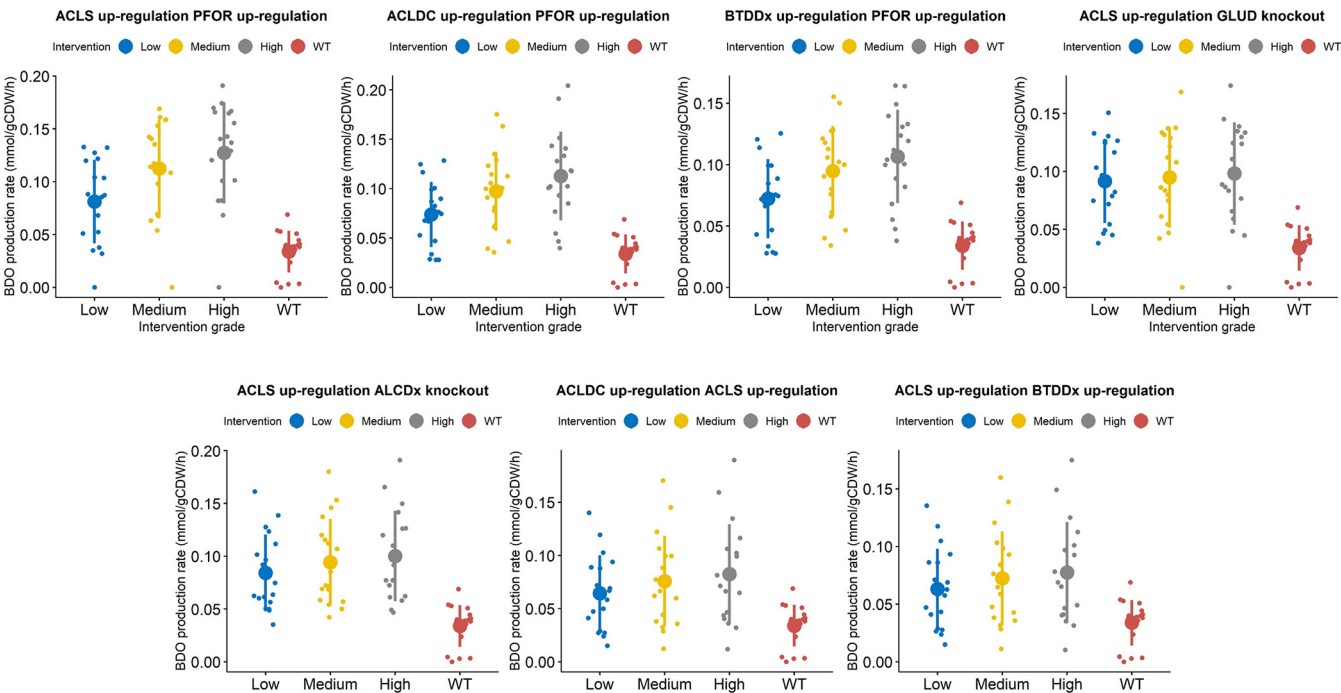

**FIG 9** Plots showing the distribution of 2,3-BDO production rates across 18 kinetic parametric sets, under the wild-type scenario and under the double-modification scenario. For each gene involved in a double intervention, three (low, medium, and high) in the levels of the enzymes catalyzing each reaction were simulated. Overlaid are the mean and standard deviation of the 2,3-BDO production rates corresponding to each intervention at each simulated change in gene expression. The figure only refers to selected double interventions reported in Table 3.

the PFOR gene alone ($flux_{BTDDx\ up,\ PFOR\ up}^{2,3-BDO} = 0.11 \pm 0.026$; $flux_{PFOR\ up,\ ACLDC\ up}^{2,3-BDO} = 0.11 \pm 0.030$). These findings highlight the benefit of combining OptForce with the kinetic ensemble model since kinetic models parameterized from experimental data can help identify the possible kinetic bottlenecks inside a metabolic pathway among the interventions suggested by OptForce that are equivalent at the stoichiometric level.

Among the double interventions that do not encompass PFOR, our analysis suggested, first, the combined overexpression of ACLS and BTDDx, and second, the overexpression of ACLS and ACLDC, which ensure an around 2-fold-higher 2,3-BDO production rate compared to the wild-type condition ($flux_{BTDDx\ up,ACLS\ up}^{2,3-BDO} = 0.073 \pm 0.030$; $flux_{ACLDC\ up,\ ACLS\ up}^{2,3-BDO} = 0.076 \pm 0.031$). ACLS and ACLDC were overexpressed individually and in combination in a patent of the Lanzatech Company (45), whereby the combined intervention allowed experimental observation of increased 2,3-BDO production in a batch culture of *C. autoethanogenum* using a CO-containing gas substrate. However, according to the outcomes of our *in silico* analysis, the double intervention did not lead to a tangible increase in the 2,3-BDO production rate compared to overexpressing exclusively the acetolactate synthase, whereas, according to reference 45, the increase in 2,3-BDO production is primarily associated with the overexpression of the acetolactate decarboxylase.

Our analysis predicted that a 2.56-fold increase in 2,3-BDO production rate compared to the wild-type case could be achievable through a double intervention, whereby the acetolactate synthase overexpression is combined with the knockout of glutamate dehydrogenase (GLUD), which catalyzes the NADPH-dependent interconversion between L-glutamate and 2-oxoglutarate ($flux_{ACLS\ up,\ GLUD\ KO}^{2,3-BDO} = 0.10 \pm 0.034$). It has been shown that *C. autoethanogenum* contains a strictly NADPH-dependent primary-secondary alcohol dehydrogenase, which could reduce acetoin to 2,3-BDO (17). Therefore, the suggested knockout of GLUD in conjunction with ACLS upregulation could favor 2,3-BDO production by avoiding NADPH consumption in the reaction catalyzed by glutamate dehydrogenase.

A slightly less effective intervention foresaw the ACLS upregulation combined with the ethanol:NAD oxidoreductase (ALCDx) knockout ($\text{flux}^{2,3-BDO}_{ACLS \text{ up, ALCDx KO}} = 0.084 \pm 0.031$). The beneficial effect on 2,3-BDO production rate is probably attributable to the negation of a competitive reaction for the electrons required for 2,3-BDO in the form of NADH.

Another double intervention featured the combination of ACLS upregulation with formate dehydrogenase (FDH) knockout ($\text{flux}^{2,3-BDO}_{ACLS \text{ up, FDH KO}} = 0.074 \pm 0.032$). Its mode of action is likely based on the manipulation of the NAD(P)H to NAD(P) ratio. Indeed, knocking out FDH affords reduced ferredoxin to build up. This excess reduced ferredoxin causes the NADP to be regenerated (reduced) to NAD(P)H, which builds an excess that must be relieved to equilibrium, and reduces acetoin to 2,3-BDO. This is supported by the fact that the flux through the NADH-dependent BTDDx increases by 26% compared to the wild-type condition.

Also, the double intervention consisting of ACLS upregulation and ACAFDOR downregulation is related to the provision of reduced ferredoxin since ACAFDOR downregulation allows limiting of the consumption of reduced ferredoxin ($\text{flux}^{2,3-BDO}_{ACLS \text{ up, ACAFDOR down}} = 0.073 \pm 0.0028$).

An alternative intervention foresaw the simultaneous upregulation of ACLS and the knockout of the rate-limiting step in gluconeogenesis catalyzed by fructose-bisphosphatase (FBP), which converts D-fructose-1,6-bisphosphate (fdp) into D-fructose-6-phosphate (f6p) and which was found to afford an almost 2-fold increase compared to the WT case ($\text{flux}^{2,3-BDO}_{ACLS \text{ up, FBP KO}} = 0.074 \pm 0.032$). Based on the changes in flux distribution observed upon this intervention, FBP knockout restricts biomass formation since fdp is rerouted from f6p to D-fructose-1-phosphate (f1p) through the action of fructose-1-phosphate kinase (FRUK). The advantage conferred to 2,3-BDO production by this intervention could derive first from the generation of ATP associated with the conversion of fdp into f1p and, second, from an increased pyruvate pool made available for 2,3-BDO through D-fructose transport via the phosphoenolpyruvate:pyruvate phosphotransferases FRUpts and FRUpts2.

**Overexpression of ACLDC on top of PFOR and ACLS predicted to be the most effective triple intervention.** When we explored the effectiveness of triple interventions, we interestingly noted that the upregulation of the entire branch from pyruvate to 2,3-BDO alone (i.e., acetolactate decarboxylase [ACLDC], acetolactate synthase [ACLS], and butanediol dehydrogenase (BTDDx]) worsens the 2,3-BDO production rate compared to overexpression of pyruvate:ferredoxin oxidoreductase alone ($\text{flux}^{2,3-BDO}_{ACLDC \text{ up, ACLS up, BTDDx up}} = 0.075 \pm 0.031$). Overexpression of BTDDx on top of the simultaneous ACLS and PFOR upregulation did not improve 2,3-BDO production compared to the double intervention ($\text{flux}^{2,3-BDO}_{ACLS \text{ up, BTDDx up, PFOR up}} = 0.14 \pm 0.029$). On the other hand, the upregulation of ACLDC on top of that of PFOR and ACLS resulted in the most effective triple intervention, further boosting the 2,3-BDO production rate by about 7% with respect to the combined PFOR and ACLS upregulation and leading to an almost 4-fold increase in 2,3-BDO production compared to the wild-type condition ($\text{flux}^{2,3-BDO}_{ACLDC \text{ up, ACLS up, PFOR up}} = 0.15 \pm 0.056$). A strain overexpressing all the three genes was characterized in continuous cultures grown on a CO-containing gaseous substrate and consistently produced higher 2,3-BDO levels compared to the plasmid control strain (38).

Additionally, our analysis pointed out triple interventions that involved ACLDC and ACLS upregulation but which, overall, were less effective than the double intervention to increase the 2,3-BDO production rate (Fig. 10). One such intervention includes the simultaneous upregulation of acetolactate synthase and acetolactate decarboxylase together with acetaldehyde:ferredoxin oxidoreductase (ACAFDOR), with a 1,86-fold increase in 2,3-BDO production with respect to the wild-type condition ($\text{flux}^{2,3-BDO}_{ACLDC \text{ up, ACLS up, ACAFDOR up}} = 0.070 \pm 0.029$). The latter enzyme, reducing undissociated acetic acid to acetaldehyde, plays a significant role in energy generation, reconstitution of oxidized ferredoxin needed in the Wood-Ljungdahl pathway, and regulation of intracellular acetate levels. According to the kinetic ensemble model used, the acetaldehyde overproduction resulting from the acetaldeyhe:ferredoxin oxidoreductase upregulation could be converted back to acetyl-CoA by the bifunctional aldehyde/alcohol

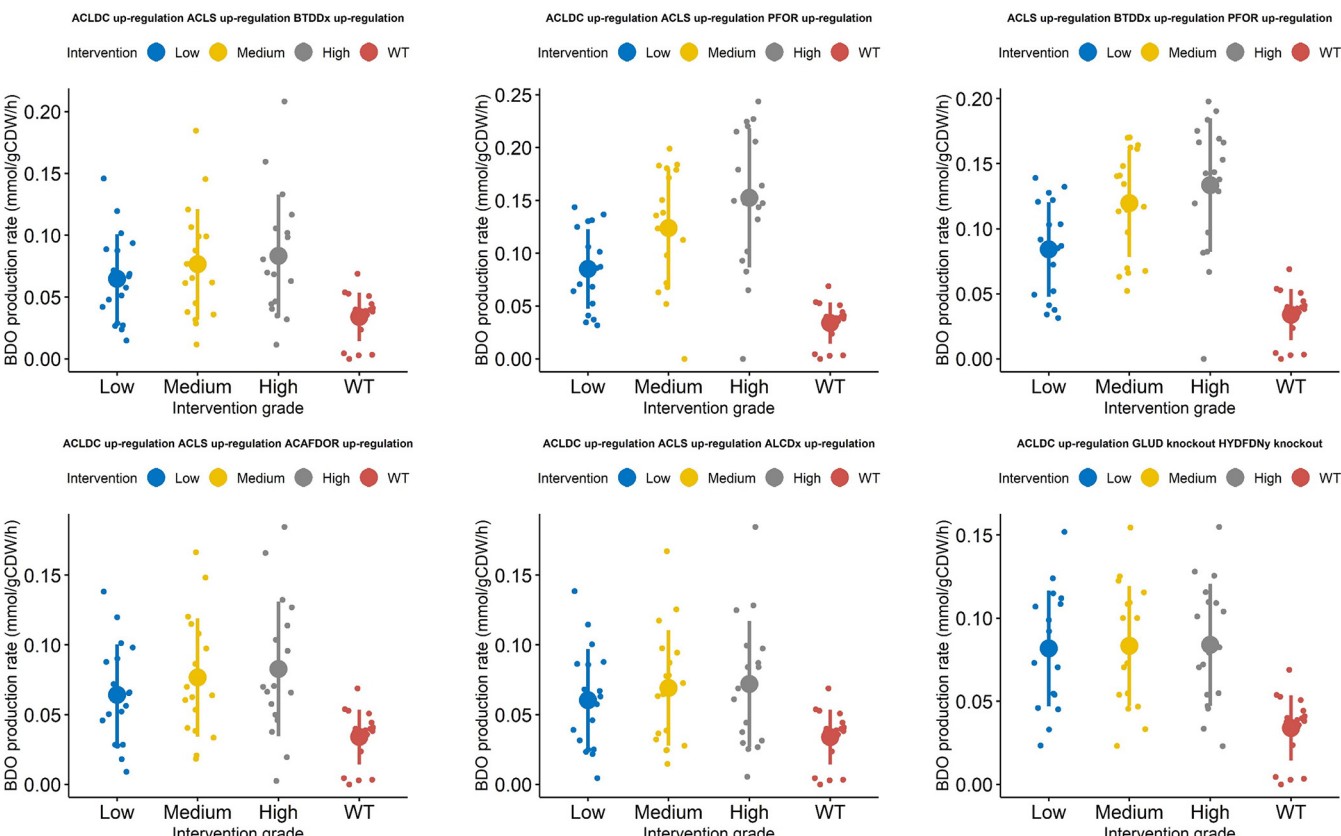

**FIG 10** Plots showing the distribution of 2,3-BDO production rates across 18 kinetic parametric sets, under the wild-type scenario and under the triple-modification scenario. For each gene involved in a triple intervention, three (low, medium, and high) in the levels of the enzymes catalyzing each reaction were simulated. Overlaid are the mean and standard deviation of the 2,3-BDO production rates corresponding to each intervention at each simulated change in gene expression. The figure only refers to selected triple interventions reported in Table 4.

dehydrogenase (NADPH-dependent ACALDy). Several lines of evidence actually suggest that the reaction catalyzed by the latter enzyme does not operate predominantly in the ethanol production direction (39, 43, 47). The Gibbs free energy estimations of the study developing the ensemble kinetic model used here (39) validated the possibility for the reaction to run in reverse (Fig. 2). Furthermore, the thermodynamics-based metabolic flux analysis of syngas chemostat cultures of *C. autoethanogenum* in reference 43 showed that the reaction catalyzed by the bifunctional aldehyde/alcohol dehydrogenase is thermodynamically prevented from operating toward ethanol formation. Furthermore, the knockout of the genes encoding the bifunctional aldehyde/alcohol dehydrogenase was found to result in growth reduction and increase in ethanol production (47). Therefore, it is plausible to hypothesize that the acetaldehyde:ferredoxin oxidoreductase, whose upregulation has been suggested in our analysis, could form an ATP-generating loop, together with the bifunctional aldehyde/alcohol dehydrogenase operating toward acetyl-CoA, the phosphate acetyltransferase and the acetate kinase.

Combining the upregulation of ACLDC and ACLS with the upregulation of either phosphate acetyltransferase (PTA), ferredoxin:NAD oxidoreductase (RNF), or acetate kinase (ACK) afforded only lower 2,3-BDO production rate compared to the double intervention encompassing the combined upregulation of ACLDC and ACLS ($\text{flux}^{2,3-BDO}_{\text{ACLDC up, ACLS up, PTA up}} = 0.050 \pm 0.026$; $\text{flux}^{2,3-BDO}_{\text{ACLDC up, ACLS ACLDC, ACLS up, RNF up}} = 0.056 \pm 0.026$; $\text{flux}^{2,3-BDO}_{\text{ACLDC up, ACLS up, ACK up}} = 0.067 \pm 0.038$).

A triple intervention leading to a 2-fold increase in 2,3-BDO production rate with respect to the wild-type case ($\text{flux}^{2,3-BDO}_{\text{ACLDC up, GLUD KO, HYDFDNy KO}} = 0.081 \pm 0.029$) consisted of the ACLDC upregulation combined with the knockout of glutamate dehydrogenase (GLUD) and the knockout of the electron-bifurcating, NADP- and ferredoxin-dependent [FeFe]-hydrogenase (HYDFDNy), which catalyzes the reversible reduction of NADP and oxidized

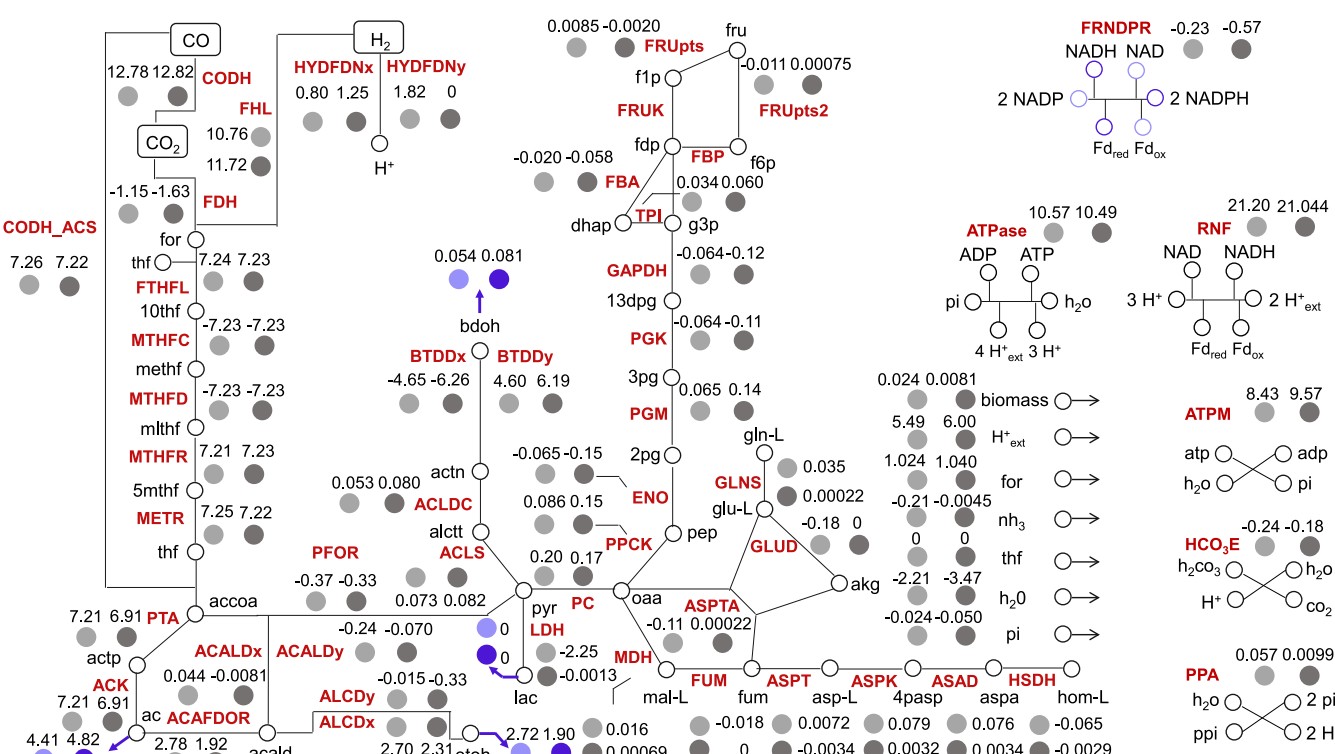

**FIG 11** Inferred changes in metabolic fluxes upon the intervention consisting of the acetolactate decarboxylase upregulation combined with the knockout of glutamate dehydrogenase and the knockout of the electron-bifurcating, NADP- and ferredoxin-dependent [FeFe]-hydrogenase. The map shows simulated fluxes for the reactions in the metabolic network representative of the core metabolism of *C. autoethanogenum* corresponding to the wild-type case (light gray) or to the triple intervention consisting of ACLDC upregulation, GLUD knockout, and HYDFDNy knockout (dark gray). Bold arrows denote by-products' exchange reactions. Reactions along with enzyme and metabolite abbreviations are described in Supplementary Table 1.

ferredoxin with 2 H₂ (Fig. 11). As regards the suggested hydrogenase knockout, it has to be noted that, in addition to the NADP-specific electron-bifurcating [FeFe]-hydrogenase, which is predominantly active on CO (16), the genome of *C. autoethanogenum* harbors also an NADH-dependent electron-bifurcating [Fe-Fe]-hydrogenase reaction, catalyzed by CAETHG_1576-78, which is reflected in the metabolic network used (15). The disruption of the NADP- and ferredoxin-dependent [FeFe]-hydrogenase severely impaired *C. autoethanogenum* growth on a CO-rich gas mix (48). However, this effect was not observable when *C. autoethanogenum* was grown on an H₂-rich gas mix since the disruption of the NADP- and ferredoxin-dependent [FeFe]-hydrogenase was compensated for by the expression of other hydrogenases (48). Several lines of evidence suggest that the compensatory role is enacted by the NADH-dependent electron-bifurcating [Fe-Fe]-hydrogenase. Indeed, the inactivation of this hydrogenase caused the microorganism to grow poorly under the H₂-rich condition (48). Moreover, transcriptomic data of a steady-state culture of *C. autoethanogenum*, grown on a H₂-rich gas mix over 23 days, showed that the expression of the NADH-dependent electron-bifurcating [Fe-Fe]-hydrogenase increased progressively to reach similar levels of expression as the NADPH-dependent electron-bifurcating [Fe-Fe]-hydrogenase (48). Interestingly, the expression of this hydrogenase did not change in *C. autoethanogenum* fermenting a CO-rich gas mix (23). The increased expression under the H₂-rich condition suggests that the NADH-dependent electron-bifurcating [Fe-Fe]-hydrogenase plays a critical role in H₂ uptake. Notably, the mutant with inactivated NADP- and ferredoxin-dependent [FeFe]-hydrogenase produced greater amounts of reduced by-products, such as ethanol, compared to the wild-type strain under the H₂-rich condition (48).

The reduction of acetoin to 2,3-BDO can be carried out by a butanediol dehydrogenase (CAETHG_0385), whose activity has been shown to be favored by NADH (5). Therefore, it is plausible that the reducing equivalents generated by the NAD- and ferredoxin-dependent

[FeFe]-hydrogenase, which was shown able to compensate for the disruption of the NADP- and ferredoxin-dependent [FeFe]-hydrogenase in an $H_2$-rich gas mix, could be used by the NADH-dependent butanediol dehydrogenase to fix carbon into 2,3-BDO.

Finally, the primary-secondary NADPH-alcohol dehydrogenase, which is encoded by CAETHG_0553 in *C. autoethanogenum*, has been shown to be involved in ethanol (18) and 2,3-BDO (16, 17) synthesis. Therefore, it is conceivable that the knockout of the NADPH-dependent glutamate dehydrogenase, foreseen in the triple intervention, could increase the availability of reducing equivalents for the 2,3-BDO synthesis by the primary-secondary NADPH-alcohol dehydrogenase.

Overall, our analysis allowed us to identify and comparatively assess the effect on 2,3-BDO production rate of a wide range of single, double, and triple interventions. Intuitive and experimentally pursued metabolic engineering strategies, namely, the single pyruvate oxidoreductase overexpression or the combined overexpression of pyruvate oxidoreductase with the downstream acetolactate synthase and acetolactate decarboxylase enzymes, were reliably recapitulated (45).

Interestingly, our analysis could prioritize single, double, and triple interventions by the relative increase ensured by the 2,3-BDO production rate. For instance, our analysis, thanks to the integrated usage of OptForce and kinetic modeling, identified PFOR as the bottleneck reaction in the pathway leading from acetyl-CoA to 2,3-BDO. Indeed, only the combined upregulation of PFOR with ACLS, among the double interventions, and the combined upregulation of PFOR with ACLS and ACLDC, among the triple interventions, conferred an increment in 2,3-BDO production rate compared to the PFOR upregulation alone.

Noteworthy, according to our analysis, interventions partially impinging on by-products branching from acetyl-CoA and pyruvate (acetate, ethanol, amino acids) offer valuable alternatives to the interventions focusing directly on the specific branch from pyruvate to 2,3-BDO. Among the alternative options, the ACK downregulation, the double interventions combining ACLS upregulation with glutamate dehydrogenase or ethanol:NAD oxidoreductase knockout, and the triple intervention consisting of the ALCDC upregulation with the knockouts of glutamate dehydrogenase and electron-bifurcating, NADP- and ferredoxin-dependent [FeFe]-hydrogenase are worthy of attention and further exploration.

## MATERIALS AND METHODS

**Genome-scale stoichiometric metabolic model.** A couple of genome-scale metabolic models (GEMs) are publicly available for *C. autoethanogenum*. The *C. autoethanogenum* GEM iCLAU786 was realized in reference 23 and refined in reference 24. The *C. autoethanogenum* GEM Metaclau was made available in reference 25. iCLAU786 contains 1,094 metabolites, 1,108 reactions, and 643 genes, whereas Metaclau contains 855 metabolites, 849 reactions, and 532 genes. Prior to using GEMs for *in silico* design of strains overproducing 2,3-BDO, we validated each GEM by running three tests. We assessed (i) the flux through the network in the absence of flux through the exchange reactions, and (ii) the flux through the non-growth-associated ATP hydrolysis reaction in the absence of flux through the uptake reactions. (iii) Finally, we benchmarked condition-specific models against several experimental data sets. Step-by-step benchmarking of both models is provided in Validation.m MATLAB script in the GitHub repository (https://github.com/chan-csu/OptForce_Bdoh/Report/ModelValidation/Validation.m).

**Flux through the network with zero exchanges.** Exchange reactions were found in iCLAU786 and Metaclau, fluxes through exchange reactions were set to zero, and the sum of absolute flux through the whole network was maximized.

**Flux through ATP hydrolysis reaction with zero uptakes.** Uptake reactions were found in iCLAU786 and Metaclau, fluxes through exchange reactions were set to zero, and the ATP hydrolysis flux was maximized (49).

**Condition-specific model validation.** The flux variability analysis (FVA) technique uses two linear optimization problems for each reaction of a genomic-scale metabolic reconstruction to evaluate the minimum and maximum values of each reaction rate satisfying the constraints, which occur as inequalities imposing bounds on the system and as mass balance equations imposing mass conservation at steady state. Results obtained through the application of FVA provide insight into the capabilities of a metabolic network to investigate the steady-state behavior of a microorganism. Before applying the model for *in silico* design of strains with potentially superior characteristics, it is advisable to assess the accuracy of the model by constraining the GEM with publicly available experimental data. More precisely, we constrained the GEM with substrate uptake rates, maximized biomass yield in FVA calculations, and compared model predictions with published experimental data concerning specific growth rate and by-products' production rates. We surveyed the literature stored at Web of Science Core Collection, PubMed, and individual publisher websites for research articles

focused on CO-based gas fermentations with *C. autoethanogenum* and reporting quantitative data on gas exchange rates, as well as biomass and by-products' formation rates (Table 1). FVA as a means of model validation was run using scripts in the Cobra Toolbox v.3.0 (42) and Gurobi Optimizer version 9.1 (https://www.gurobi.com/products/gurobi-optimizer/) as the optimization solver. The GEM was constrained with experimentally quantitated gas uptake rates. The objective used in the simulations was maximizing the biomass formation rate. Finally, the experimentally reported values for acetate, ethanol, 2,3-BDO, and the biomass formation rate were contrasted with the range of allowed values resulting from FVA. Furthermore, flux balance analysis (FBA) was used to predict acetate, ethanol, 2,3-BDO, and the biomass formation rate corresponding to each data set included in Table 1 and to assess the consistency of FBA predicted values with the experimental values reported in Table 1. The files containing the experimental data obtained by literature mining, the iCLUA786 and Metaclau GEMs, and the script used for model validation are available in the GitHub repository (https://github.com/chan-csu/OptForce_Bdoh/tree/master/Report/Model_Validation).

**Relationship between gas feeding and $CO_2$ production.** We varied the fluxes through CO and $H_2$ uptake reactions stepwise in the range from 0 mmol/$g_{CDW}$/h to 50 mmol/$g_{CDW}$/h, and the optimal objective value was calculated as a function of those fluxes. We investigated the effects of various gaseous substrates' uptake rates on 2,3-BDO production rate and $CO_2$ production rate.

**Prediction of interventions enhancing 2,3-butanediol production.** The genome-scale metabolic reconstruction Metaclau was constrained with CO and $H_2$ uptake rates set to 20 and 33 mmol/$g_{CDW}$/h, which we retrieved from the gas fermentation study described in reference 40. The model was used to simulate the *in silico* design and screening of strains with gene modifications which, subject to the model stoichiometry and boundary conditions, could lead to the production of higher yields of the 2,3-BDO compound. The design and screening of mutant strains were carried out using the computational tool OptForce (41) as it is implemented in the Cobra Toolbox v.3.0 within the MATLAB environment. In the first step, the biomass exchange reaction was set as the objective function to find the maximum attainable growth rate, and this was assumed to present the wild-type strain. The maximum attainable growth rate was multiplied by 0.1 to obtain a lower bound on growth to ensure that the organism can grow in all of the suggested interventions. In the next step, OptForce with 2,3-BDO exchange as objective was used to find 2,3-BDO maximizing phenotype. OptForce compares the flux ranges between the wild-type and butanediol-maximizing mutant by using flux variability analysis to suggest interventions (https://github.com/chan-csu/OptForce_Bdoh/blob/master/Final_Script.m). The perturbations suggested by OptForce to meet the overproduction target can include the increase or decrease in the flux values of particular reactions or the elimination of particular reaction fluxes, which can be achievable through genes' overexpression, downregulation, or knockout, respectively. In particular, the top 500 theoretical interventions (first-, second-, and third-order interventions), which were found consistent with the imposed 2,3-BDO overproduction target according to the OtpForce strain design procedure, were retained for further independent validation.

**Simulations using the kinetic model of *C. autoethanogenum*.** Interventions were further analyzed through a kinetic model of the *C. autoethanogenum* core metabolism previously developed with the ensemble modeling framework (39). The kinetic model accounts for 70 reactions and 62 metabolites. Briefly, in the ensemble modeling framework, every reaction in the network was decomposed as elementary reactions with elementary kinetics. Using the Gibbs free energy and range for internal metabolite concentrations as constraints, a large number of thermodynamically feasible kinetic parameter sets were randomly sampled and screened for the convergence and fitness to three sets of experimental growth data for *C. autoethanogenum* in reference 39, finally resulting in an ensemble of 18 kinetic parameter sets that were used in this work to validate OptForce predictions. The validation of OptForce predictions by kinetic modeling was restricted to the subset of reactions encompassed by both methods. Table S1 on GitHub provides the lists of reactions and metabolites included in the metabolic reconstruction (https://github.com/chan-csu/OptForce_Bdoh/blob/59a42acaa7be018702cd748a441a6076391306c6/Results/Supplementary_Table_1.xlsx). The kinetic ensemble model of *C. autoethanogenum* was employed to calculate the change in 2,3-BDO production rate as a function of changing enzymes' expression levels according to the combinations of single, double, and triple interventions on reactions' fluxes, which were suggested by OptForce. We set three fold change (FC) values in enzymes' expression: FC = 1.1, 1.3, and 1.5 for upregulation and FC = 0.9, 0.7 and 0.5 for downregulation. The kinetic representation of the core metabolism was constrained by a single reference state where the CO and $H_2$ uptake rates were set to 20 and 33 mmol/$g_{CDW}$/h, respectively. By calculating the forward finite difference slopes for each intervention across each of the final 18 kinetic parameter sets, we obtained a distribution of 2,3-BDO production rates in correspondence to each intervention. In order to identify the interventions that reflect a statistically significant higher production of 2,3-BDO compared to the wild-type strain, we applied the pairwise Wilcoxon's test. Raw *P* values were adjusted for multiple testing by the Bonferroni approach. Statistical analysis was performed within the R software environment for statistical computing.

## SUPPLEMENTAL MATERIAL

Supplemental material is available online only.

**DATA SET S1**, XLSX file, 0.02 MB.
**FIG S1**, TIF file, 0.2 MB.
**FIG S2**, TIF file, 0.4 MB.
**FIG S3**, TIF file, 0.3 MB.
**FIG S4**, TIF file, 0.3 MB.
**FIG S5**, TIF file, 0.3 MB.

**TABLE S1**, XLSX file, 0.02 MB.
**TABLE S2**, XLSX file, 0.01 MB.
**TABLE S3**, XLSX file, 0.01 MB.
**TABLE S4**, XLSX file, 0.01 MB.

## ACKNOWLEDGMENTS

This work was supported by the program POR FESR 2014/2020—Action I.1b.2.2—Bioeconomy Technological platform under grant 333-30 (project PRIME "Processi e pRodotti Innovativi di Chimica vErde"). The funders had no role in study design, data collection and interpretation, or the decision to submit the work for publication.

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
