## [Reviewer comments · mSystems]

Metabolic engineering interventions for sustainable 2,3-butanediol production in gas fermenting *Clostridium autoethanogenum*

Parsa Ghadermazi, Angela Re, Luca Ricci, and Siu Chan

Corresponding Author(s): Angela Re, Istituto Italiano di Tecnologia

Review Timeline:

Submission Date:	September 8, 2021
Editorial Decision:	November 15, 2021
Revision Received:	January 4, 2022
Accepted:	February 18, 2022

Editor: Rafael Silva-Rocha

Reviewer(s): Disclosure of reviewer identity is with reference to reviewer comments included in decision letter(s). The following individuals involved in review of your submission have agreed to reveal their identity: Wei Xiong (Reviewer #1); Yinjie J. Tang (Reviewer #2)

Transaction Report:

DOI: <https://doi.org/10.1128/mSystems.01111-21>

November 15, 2021

Dr. Angela Re
Istituto Italiano di Tecnologia
Centre for Sustainable Future Technologies
Via Livorno 60
Torino, Piemonte 10129
Italy

Re: mSystems01111-21 (Metabolic engineering interventions for sustainable 2,3-butanediol production in gas fermenting *Clostridium autoethanogenum*)

Dear Dr. Angela Re:

Thank you for submitting your manuscript to mSystems. We have completed our review and I am pleased to inform you that, in principle, we expect to accept it for publication in mSystems. However, acceptance will not be final until you have adequately addressed the reviewer comments.

Preparing Revision Guidelines

Sincerely,

Rafael Silva-Rocha

Editor, mSystems

Journals Department
Reviewer comments:

Reviewer #1 (Comments for the Author):

The manuscript by Ghadermazi et al. described a state-of-the-art computational approach to predict most likely gene targets, modifications of which could improve 23BDO productivity in a syngas-fermenting bacterium. Both the genome-scale FBA model and the kinetic model based on ensemble modeling are employed. The modeling outcomes provided actionable information to strain engineers. Given the potential value of syngas fermentation on a sustainable bioeconomy and the advances of modeling approaches on metabolic engineering, this work offered sufficient interest to the audience timely.

The reviewer located a few places where revisions will be required.

Major:

1. C. autoethanogenum core metabolism was shown in Figure 2. However, the reviewer took a while to locate the 23BDO pathway there. Considering 23BDO is a target product of this work, its pathway should be labeled much more clearly.
2. Ensemble modeling is the key innovation point of this paper. Thus, background information on this new approach should be provided more adequately. Citing previous literature is good, but not enough. For example, please describe more details about how and why the 18 parameter sets were selected in the ensemble. In addition, enzyme level interventions (low (+/-20%), medium (+/-40%) or high (+/-60%)) seem to be too small for experimental practice. What if the expression levels are up/down-regulated, for example, at most 5 folds? Did the parameter sets consider the robustness effect and/or thermodynamic constraints on the pathway? The reviewer assumes so. But more comprehensive information will help readers to understand this new modeling approach much better.

Minor:

1. Per Line 132 Model validation, iCLAU786 and Metaclau model were compared for FVA and the iCLAU786 model failed the test. This section also stated that "the accuracy in predicting the acetate, ethanol and 2,3-BDO production rates and the specific growth rate was found to be superior using Metaclau compared to iCLAU786." Possible reasons should be provided.
2. Figure 1. The blue dots and orange dots seem to be mis-denoted in the caption.
3. Why did FBA suggest no difference of specific growth rates in varied conditions in Fig.1?
4. Please consider citing the most relevant works for ensemble modeling, for example
Metabolic Engineering. 2014,25, 63-71
Metabolic Engineering. 2020,57,140-150

Reviewer #2 (Comments for the Author):

This paper used GEMs to guide Metabolic engineering interventions for Clostridium gas fermenting of 2,3-butanediol. Via flux simulations, bottle neck reactions in relevant pathways were identified and the targets offered rational genetic modification (a triple intervention) for improved biosynthesis. The overall paper is innovative and the modeling work and algorithms is of high importance for computational strain design. However, I still have a few minor suggestions for authors to clarify their work and improve their results.

1. In the introduction, the authors may add a pathway map to show the key reactions for 2,3-butanediol synthesis. This will be helpful for readers to understand pathway modifications. Also, their Fig 3 heatmap seems a bit hard to be understood. Fig 3 caption should include the names of each metabolite abbreviations.
2. The authors need to consider that acetate can serve as an important carbon source during gas fermentation. The accumulated acetate in early growth stage can be re-used for bio-production of alcohols. Can the GEMs simulate the dynamic changes between acidogenesis and solventogenesis?
3. The authors assessed GEM quality by basic tests and flux variability analysis (FVA). It is unclear to me why tests focus on internal reactions to detect thermodynamically infeasible cycles and non-growth-associated maintenance reaction? Technically, FBA based approaches are unable to determine metabolic futile cycles or maintenance costs. The measurement of metabolic cycles and energy fluxes usually requires 13C-MFA (Biotechnology and bioengineering 111 (3), 575-585). Authors should justify their tests. Also, can they use 13C-MFA results to check the GEM quality?
4. In the section of Analysis of the effect of H₂ on 2,3-butanediol and CO₂ production, The model indicates that a high-H₂/CO gas fermentation process is advantageous. What is the suggested optimal ratio by the model? The experiments showed an increase in cellular maintenance loss with increasing H₂ supply. Therefore, it must be an optimal point for H₂/CO ratio. Can the GEM model address this observation/issue?

5. In reality, The H₂ and CO uptakes are influenced by mass transports (Scientific reports 7 (1), 1-11). H₂ is more expensive and less soluble in the culture. Its uptake by cells is slower than CO. How gas mass transfer process affects the outputs of kinetic ensemble models?

6. Their findings on PFOR overexpression and an ATP-generating loop are very interesting. How this clostridium host makes ATP? Can authors provide a more detailed energy flux map to show key reactions for ATP, NADH and NADPH generations? Also, the paper can be stronger if authors add reaction thermodynamic information for these targeted reactions.

7. In the materials and Method section, I am not sure how authors set up their objective functions for FBA simulations and for OptForce. Background information could be provided. Also, kinetic models were used in the simulation. Authors needs to add some introductions to kinetic model equations and algorithms.

We provide a point-by-point responses to the issues raised by the reviewers.

Reviewer #1 (Comments for the Author):

The manuscript by Ghadermazi et al. described a state-of-the-art computational approach to predict most likely gene targets, modifications of which could improve 23BDO productivity in a syngas-fermenting bacterium. Both the genome-scale FBA model and the kinetic model based on ensemble modeling are employed. The modeling outcomes provided actionable information to strain engineers. Given the potential value of syngas fermentation on a sustainable bioeconomy and the advances of modeling approaches on metabolic engineering, this work offered sufficient interest to the audience timely.

The reviewer located a few places where revisions will be required.

Major:

1. *C. autoethanogenum* core metabolism was shown in Figure 2. However, the reviewer took a while to locate the 23BDO pathway there. Considering 23BDO is a target product of this work, its pathway should be labeled much more clearly.

We improved the visualization of the 2,3-BDO biosynthetic pathway in the revised Figure 2 by coloring the 2,3-BDO biosynthetic pathway. Furthermore, we added a new Figure 1 focusing exclusively on the 2,3-BDO biosynthetic pathway.

2. Ensemble modeling is the key innovation point of this paper. Thus, background information on this new approach should be provided more adequately. Citing previous literature is good, but not enough. For example, please describe more details about how and why the 18 parameter sets were selected in the ensemble. In addition, enzyme level interventions (low (+/-20%), medium (+/-40%) or high (+/-60%)) seem to be too small for experimental practice. What if the expression levels are up/down-regulated, for example, at most 5 folds? Did the parameter sets consider the robustness effect and/or thermodynamic constraints on the pathway? The reviewer assumes so. But more comprehensive information will help readers to understand this new modeling approach much better.

We thank the Reviewer for asking for this supplement of information that we provide herein and that we integrated in the revised version of the manuscript.

Describe more details about how and why the 18 parameter sets were selected in the ensemble.

The kinetic model currently being used was constructed in Greene et al., 2019, *Biochemical engineering journal*, 148, 46-56. Initially 359,000 sets of kinetic parameters consistent with the reference state flux distribution were sampled and iteratively tested for their convergence and fitness against other experimental conditions to distill down to finally 18 parameter sets. This ensemble of 18 kinetic models are equally capable of reproducing the experimental data, and generate thermodynamically feasible results.

We have included more details in the methods subsection titled 'Simulations using the kinetic model of *C. autoethanogenum*'

Did the parameter sets consider the robustness effect and/or thermodynamic constraints on the pathway?

During the sampling step mentioned above in Greene et al., 2019, *Biochemical engineering journal*, 148, 46-56, the authors introduced bounds for metabolite concentrations (0.01~20 mM) and constrained the parameter space to lead to feasible actual free energy for each reaction. Technically, it was done by sampling the ratio of forward to reverse fluxes at each decomposed elementary reaction step such that the product of all these ratios is bounded by the estimated range of actual Gibbs free energy (see equations 6 to 13 in Tran et al., 2008, *Biophysical journal*,

95(12), 5606-5617 for mathematical details). This way, all the parameter sets are guaranteed to generate thermodynamically feasible solutions.

What if the expression levels are up/down-regulated, for example, at most 5 folds?

Theoretically, it is possible to predict the effect of a larger perturbation. Following the reviewer's suggestion, we repeated the simulations to perturb the enzyme levels by 3, 5 and 10 folds for upregulation and 0.3, 0.2, and 0.1 fold change for downregulation (newly added Supplementary Tables 2, 3 and 4 and Supplementary Figures 3, 4 and 5 in the revised Supplementary Information). The trend obtained from these results show that the same strategies are picked as successful strategies. Therefore, we conclude that the current model is possible to predict the impact of larger enzyme fold changes.

However, we do want to note that there are more potential issues as we try to increase the perturbation issues. First, as we also observed in our simulation results, the chance for some of the parameter sets to become unstable at higher perturbation levels increases. This could be expected for many dynamical systems as the stability generally decreases when the perturbation is larger. A model parameterized using more datasets from diverse conditions and diverse enzyme expression profile might help increase the overall model stability. Second, upon a large perturbation, resource reallocation at the protein level could happen inside the cells and deviates from the kinetic model in which constant expression level for all other unperturbed enzymes is assumed. That being said, without any additional datasets and efforts to improve the model parameterization or implement mechanisms/kinetics for protein expression, we believe that the kinetic model is still a useful system modeling tool that captures existing data and predicts some scenarios under large perturbations.

We have now included a brief related discussion at the end of the subsection 'Prediction of genetic manipulations leading to 2,3-butanediol overproduction.'

Minor:

1. Per Line 132 Model validation, iCLAU786 and Metaclau model were compared for FVA and the iCLAU786 model failed the test. This section also stated that "the accuracy in predicting the acetate, ethanol and 2,3-BDO production rates and the specific growth rate was found to be superior using Metaclau compared to iCLU786." Possible reasons should be provided.

The primary reason that we believe for why Metaclau has a better accuracy in predicting product formation and growth rates than iCLAU786 is a better model consistency (e.g., mass and charge balance, absence of infeasible energy-generating cycles). As we also noted in the same paragraph before this sentence, Metaclau does not have any infeasible energy-generating cycles (internal cycles that are perfectly carbon balanced with an overall effect of converting $ADP + Pi + H$ to $ATP + H_2O$, i.e., ATP synthesis) while iCLAU786 has (https://github.com/chan-csu/OptForce_Bdoh). This means that during FBA simulations, the ATP production in iCLAU786 is actually unlimited and not bounded by the uptake conditions. But we cannot rule out other possible reasons such as better annotation in one model versus the other, or a more accurate biomass reaction stoichiometry. A detailed model comparison, which is outside the scope of this study, is needed to find out the true underlying reasons.

2. Figure 1. The blue dots and orange dots seem to be mis-denoted in the caption.

3. Why did FBA suggest no difference of specific growth rates in varied conditions in Fig.1?

We thank the Reviewer for noticing the error in the figure. The colors corresponding to experimental and FBA-derived values were erroneously assigned. Due to this error, the specific growth rate appeared unchanged across various conditions. This mistake was amended in the current version of Figure 3.

4. Please consider citing the most relevant works for ensemble modeling, for example
Metabolic Engineering. 2014,25, 63-71
Metabolic Engineering. 2020,57,140-150

We thank the Reviewer for suggesting these references that were employed in the revised version of the Introduction section when we expanded the introduction to the kinetic modeling, and the kinetic ensemble modeling in particular.

Reviewer #2 (Comments for the Author):

This paper used GEMs to guide Metabolic engineering interventions for *Clostridium* gas fermenting of 2,3-butanediol. Via flux simulations, bottle neck reactions in relevant pathways were identified and the targets offered rational genetic modification (a triple intervention) for improved biosynthesis. The overall paper is innovative and the modeling work and algorithms is of high importance for computational strain design. However, I still have a few minor suggestions for authors to clarify their work and improve their results.

1. In the introduction, the authors may add a pathway map to show the key reactions for 2,3-butanediol synthesis. This will be helpful for readers to understand pathway modifications. Also, their Fig 3 heatmap seems a bit hard to be understood. Fig 3 caption should include the names of each metabolite abbreviations.

Actions undertaken:

We thank the Reviewer for his/her suggestion and in the revised version of the manuscript we added the present Figure 1 showing the 2,3-butanediol biosynthetic pathway. Furthermore, we highlighted the 2,3-butanediol biosynthetic pathway in the overall metabolic network shown in Figure 4.

We agree that row labels in the heatmap are not self-explaining. However, adding full names in the caption is impractical. Therefore, we refer the reader to the Supplementary Table 1 of the manuscript where both reactions and metabolites are fully detailed.

2. The authors need to consider that acetate can serve as an important carbon source during gas fermentation. The accumulated acetate in early growth stage can be re-used for bio-production of alcohols. Can the GEMs simulate the dynamic changes between acidogenesis and solventogenesis?

For the kinetic model and the associated parameter sets currently being used, they are trained from data where CO and H₂ are being consumed. In our testing, setting other substrates as uptake such as acetate would make the model unstable. Related to the response to Reviewer 1's major comment #2, re-parameterizing the model by incorporating physiological and gene expression data under acetate consuming conditions might help capture this. Alternatively, in the absence of additional data, performing dynamic flux balance analysis (dFBA) using the genome-scale metabolic models for *C. autoethanogenum* could probably predict that since the genome-scale model is capable of predicting the effect of acetate uptake. While we thank the reviewer for this constructive idea and agree that this is an exciting area that can be further explored in the future for optimizing the entire fermentation, this is beyond the scope of this current paper. Our goal is more revolving around identifying the genetic targets for improving butanediol production from CO fermentation with Hydrogen.

3. The authors assessed GEM quality by basic tests and flux variability analysis (FVA). It is unclear to me why tests focus on internal reactions to detect thermodynamically infeasible cycles and non-growth-associated maintenance reaction? Technically, FBA based approaches are unable to determine metabolic futile cycles or maintenance costs. The measurement of metabolic cycles and energy fluxes usually requires ¹³C-MFA (Biotechnology and bioengineering 111 (3), 575-585). Authors should justify their tests. Also, can they use ¹³C-MFA results to check the GEM quality?

We agree with the Reviewer that intracellular metabolic cycles and energy fluxes need ¹³C-MFA data to identify. It will be great to have such a dataset for this organism except that currently the data available is very limited. For the model consistency test, we would like to point out that thermodynamically infeasible cycles (TICs) are different from futile cycles. Futile cycles happen in real biological systems and just consume energy (into heat for example), but TICs are computationally artifacts in metabolic models that theoretically should not happen in the system. We want to avoid them especially for the TICs that can generate energy out of nothing (instead of consuming energy like futile cycles) (Fritzemeier et al. 2019, *PLoS Comput Biol* 13(4): e1005494). The net effect of those TICs is to generate energy ($\text{ADP} + \text{P}_i + \text{H} \rightarrow \text{ATP} + \text{H}_2\text{O}$) internally without consuming any carbon source or cofactor, so any energy (ATP) constraint on the model, e.g., NGAM, or the ATP required for synthesizing any metabolites, or the biomass, can be arbitrarily satisfied by those TICs, rendering incorrect results. To detect those cycles efficiently, a simple method that we used is to shut down all exchange fluxes and find any internal cycles that can be coupled to the ATP hydrolysis reaction ($\text{ATP} + \text{H}_2\text{O} \rightarrow \text{ADP} + \text{P}_i + \text{H}$) which is also usually used to model the non-growth associated maintenance (NGAM) when predicting growth by FBA for example..

4. In the section of Analysis of the effect of H₂ on 2,3-butanediol and CO₂ production, The model indicates that a high-H₂/CO gas fermentation process is advantageous. What is the suggested optimal ratio by the model? The experiments showed an increase in cellular maintenance loss with increasing H₂ supply. Therefore, it must be an optimal point for H₂/CO ratio. Can the GEM model address this observation/issue?

To test what the Reviewer suggested, we ran another simulation where we held CO uptake constant and varied H₂ uptake (see the attached figure which corresponds to supplementary Figure 2).

We did observe a decrease in the BDO yield per hydrogen uptake. Especially after the uptake rate of hydrogen is > 20 mmol/gDW/hr, no increase in BDO or any other significant product profile change is observed. And the Reviewer was correct about maintenance in the sense that we observed that the additional electrons at high hydrogen uptake are consumed in some futile cycles in the model. The primary reason for this model behavior is that all kinetics in the model essentially follow Michaelis-Menten kinetics and the hydrogen-consuming reactions that lead to product formation will ultimately get saturated at a certain hydrogen concentration that is correlated to the H_2 uptake rate.

In this sense, the kinetic model could suggest the optimal H_2/CO ratio depending on the optimality criteria (e.g., BDO yield alone, or considering also the cost of hydrogen, undesired CO_2 release). The bottom line is that at a certain CO uptake rate, the model can predict a maximum H_2 uptake rate beyond which further H_2 uptake does not improve the process.

We have included the discussion of these results in the subsection titled 'Analysis of the effect of H_2 on 2,3-butanediol and CO_2 production' in the Results section.

5. In reality, The H_2 and CO uptakes are influenced by mass transports (Scientific reports 7 (1), 1-11). H_2 is more expensive and less soluble in the culture. Its uptake by cells is slower than CO . How gas mass transfer process affects the outputs of kinetic ensemble models?

We thank the Reviewer for pointing to this. All the uptake fluxes that we used in this analysis are taken from experimentally observed values. Both flux variability analysis and the kinetic model simulations show that varying H_2 uptake while holding CO uptake flux constant, will not affect the simulations' results beyond a certain point ~ 50 mmol/gDWh. This shows that mass transfer limitations will not affect our simulation results. This is partly because of enzyme saturation in the kinetic model, and partly because of the stoichiometry embedded in the genome-scale metabolic model. The mass transfer limitations will limit the maximum uptake flux values determined for the kinetic model, and since our values are from the experimentally observed values, we are not concerned about passing the mass transfer limit. To explicitly incorporate the reactor kinetics, a future possibility is to embed FBA/kinetic models into reactor kinetics similar to Chen et al. (2018, Biochem Eng J 129: 64-73).

6. Their findings on PFOR overexpression and an ATP-generating loop are very interesting. How this clostridium host makes ATP? Can authors provide a more detailed energy flux map to show key reactions for ATP, NADH and NADPH generations? Also, the paper can be stronger if authors add reaction thermodynamic information for these targeted reactions.

Actions undertaken:

We thank the Reviewer for his/her observation. We described the key reactions supporting energetics and redox factor generation in acetogens of the genus Clostridium in the Introduction section and in the present Figure 2. Furthermore, we included information on DrG'mmin and DrG'mmax according to the paper presenting the ensemble kinetic model. Finally, when discussing the intervention foreseeing the ACLS, ACLDC and ACAFDOR up-regulation, we highlighted that the interpretation of this intervention is consistent with the thermodynamic constraints to which the involved reactions are bound.

7. In the materials and Method section, I am not sure how authors set up their objective functions for FBA simulations and for OptForce. Background information could be provided. Also, kinetic models were used in the simulation. Authors needs to add some introductions to kinetic model equations and algorithms.

In the first step, we used the experimental dataset taken from Valgepea et. al 2018. Then, we maximized the biomass reaction to find the maximum attainable growth rate, and we used this phenotype as the wild type as it is a common practice to assume the wildtype is a phenotype that maximizes the biomass production. We used one tenth of this value as a lower bound on growth to ensure that the organism can grow in all of the suggested interventions. In the next step we used Optforce with BDO as the objective function. OptForce compares the flux ranges between the wildtype and BDO maximizing mutant using flux variability analysis to suggest interventions, as described in Ranganathan et al., (2010, PLOS Computational Biology 6(4): e1000744). This resulted in a list of first, second, and third order interventions that we mapped to the kinetic model. The matlab script used for this process has been provided in the GitHub repository for this project. (https://github.com/chan-csu/OptForce_Bdoh/blob/master/Final_Script.m)

Some background information about the ensembling kinetic modeling framework is added to the Materials and Methods section in the revised manuscript:

Briefly, in the ensemble modeling framework, every reaction in the network was decomposed as elementary reactions with elementary kinetics. Using the Gibbs free energy and range for internal metabolite concentrations as constraints, a large number of thermodynamically feasible kinetic parameter sets were randomly sampled and screened for the convergence and fitness to three sets of experimental growth data for C. autoethanogenum in Greene et al. 2019 (42), finally resulting in an ensemble of 18 kinetic parameter sets that were used in this work to validate Optforce predictions.

February 18, 2022

Dr. Angela Re
Istituto Italiano di Tecnologia
Centre for Sustainable Future Technologies
Via Livorno 60
Torino, Piemonte 10129
Italy

Re: mSystems01111-21R1 (Metabolic engineering interventions for sustainable 2,3-butanediol production in gas fermenting *Clostridium autoethanogenum*)

Dear Dr. Angela Re:

Your manuscript has been accepted, and I am forwarding it to the ASM Journals Department for publication. For your reference, ASM Journals' address is given below. Before it can be scheduled for publication, your manuscript will be checked by the mSystems production staff to make sure that all elements meet the technical requirements for publication. They will contact you if anything needs to be revised before copyediting and production can begin. Otherwise, you will be notified when your proofs are ready to be viewed.

Publication Fees:

We recognize that the video files can become quite large, and so to avoid quality loss ASM suggests sending the video file via <https://www.wetransfer.com/>. When you have a final version of the video and the still ready to share, please send it to mSystems staff at mssystemsjournal@msubmit.net.

For mSystems research articles, if you would like to submit an image for consideration as the Featured Image for an issue, please contact mSystems staff at mssystemsjournal@msubmit.net.

Sincerely,

Rafael Silva-Rocha
Editor, mSystems

Journals Department
Supplemental Figure 3: Accept
Supplemental Table 3: Accept
Supplemental Figure 4: Accept
Supplemental Table 4: Accept
Supplemental Figure 5: Accept
Supplemental Table 2: Accept
Supplemental Table 1: Accept
Supplemental Figure 2: Accept
Supplemental Figure 1: Accept
Supplemental File 1: Accept